

# Cloud and aerosol radiative effects as key players for anthropogenic changes in atmospheric dynamics over southern West Africa

Konrad Deetz[1], Heike Vogel[1], Peter Knippertz[1], Bianca Adler[1], Jonathan Taylor[2], Hugh Coe[2], Keith Bower[2], Sophie Haslett[2], Michael Flynn[2], James Dorsey[2], Ian Crawford[2], Christoph Kottmeier[1], and Bernhard Vogel[1]

[1]Institute of Meteorology and Climate Research, Karlsruhe Institute of Technology (KIT), Karlsruhe, Germany
[2]National Centre for Atmospheric Science, and School of Earth and Environmental Sciences, University of Manchester, Manchester, United Kingdom

*Correspondence to:* Konrad Deetz (konrad.deetz@kit.edu)

**Abstract.** Southern West Africa (SWA) undergoes rapid and significant socioeconomic changes associated with a massive increase in air pollution. Still, the impact of atmospheric pollutants, in particular that of aerosol particles, on weather and climate in this region is virtually unexplored. In this study, the regional-scale model framework COSMO-ART is applied to SWA for a summer monsoon process study on 2–3 July 2016 to assess the aerosol direct and indirect effect on clouds and the

atmospheric dynamics. The modeling study is supported by observational data obtained during the extensive field campaign of the project DACCIWA (*Dynamics-Aerosol-Chemistry-Cloud Interactions in West Africa*) in June–July 2016. As indicated in previous studies, a coastal front is observed that develops during daytime and propagates inland in the evening (Atlantic Inflow). Increasing the aerosol amount in COSMO-ART leads to reduced propagation velocities with frontal displacements of 10-30 km and a weakening of the nocturnal low-level jet. This is related to a subtle balance of processes related to the decrease

in near-surface heating: (1) flow deceleration due to reduced land-sea temperature contrast and thus local pressure gradient, (2) reduced turbulence favoring frontal advance inland and (3) delayed stratus-to-cumulus transition of 1-2 h via a later onset of the convective boundary layer. The spatial shift of the Atlantic Inflow and the temporal shift of the stratus-to-cumulus transition are synergized in a new conceptual model. We hypothesize a negative feedback of the stratus-to-cumulus transition on the Atlantic Inflow with increased aerosol. The results exhibit radiation as the key player governing the aerosol affects on SWA

atmospheric dynamics via the aerosol direct effect and the Twomey effect, whereas impacts on precipitation are small.

## 1 Introduction

Atmospheric aerosol particles are highly relevant in terms of weather, climate and human health. They modify the formation of clouds and precipitation, alter the global radiation budget by scattering and absorption and can have adverse effects on the human respiratory system. The globally accelerated industrialization and urbanization are linked with increased emissions of

anthropogenic pollutants, in particular in developing and newly industrializing countries.

Southern West Africa (SWA) is densely populated and affected by land use changes and global climate change. More than half of the global population growth between now and 2050 will occur in Africa. For Nigeria, which has a population of 182 million





in 2015 (rank 7), a population increase to 399 million (rank 3) is expected for 2050 (UNO, 2015). Based on these projections, Liousse et al. (2014) show that African anthropogenic emissions will significantly increase from 2005 to 2030, if no emission regulations are implemented. The atmospheric composition over SWA is marked by a superposition of local emissions and emissions from remote areas affecting SWA through long-range transports (in particular biomass burning pollutants, Mari et al., 2008). Emissions of mineral dust, sea salt, biomass burning pollutants, biogenic volatile organic compounds (BVOCs) and anthropogenic emissions from cities and industries with the special case of gas flaring from oil industries play a role. Knippertz et al. (2017) emphasize the complexity of these anthropogenic emissions resulting i.a. from transportation, wood and waste burning or charcoal production. Hopkins et al. (2009) estimate CO, $NO_x$ and volatile organic compound emissions of the SWA megacity Lagos from aircraft measurements to be 1.44, 0.03 and 0.37 Mt $yr^{-1}$, respectively. Bahino et al. (2017) highlight the relevance of the emissions from domestic fires, with significantly increased $NH_3$ concentrations as well as traffic and industry.

We know very little about possible impacts on the regional meteorology, partly related to shortcomings in adequate observations. To overcome these shortcomings, the project *Dynamics-Aerosol-Chemistry-Cloud Interactions in West Africa* (DAC-CIWA, Knippertz et al., 2015) follows a combined observational and modeling effort for SWA. A comprehensive field campaign took place in June–July 2016 including extensive ground-based (Kalthoff et al., 2017) and airborne measurements (Flamant et al., 2018).

With respect to clouds, SWA is characterized by frequent nocturnal low-level stratus (NLLS) and stratocumulus (e.g. Schrage and Fink, 2012; Schuster et al., 2013; van der Linden et al., 2015; Adler et al., 2017) that have a significant influence on the radiation budget (e.g. Hill et al., 2017). How sensitive the cloud radiative properties react to high aerosol loadings has not been quantified. The modeling study of Lau et al. (2017) focuses on the impacts of aerosol-monsoon interactions on the variability over the northern Indian Himalaya Foothills during the summer of 2008. They highlight that the Aerosol Direct Effect (ADE), i.a. mineral dust transport and radiative heating-induced dynamical feedback processes, have major impacts on the large-scale monsoon circulation. The ADE leads to an increased north-south temperature gradient, a northward displacement of monsoon precipitation and an advanced monsoon onset over the Himalaya Foothills. The mineral dust leads to an increase in atmospheric stability via the aerosol semi-direct effect, whereupon the Aerosol Indirect Effect (AIE) may further enhance ADE by the convective-cloud invigoration mechanism (Rosenfeld et al., 2008). Lau et al. (2017) underline the need to consider aerosol-monsoon interactions even in short-term numerical forecasting of the monsoon circulation and precipitation.

In a modeling study of marine warm-cloud regimes, Saleeby et al. (2014) show that an increase in the amount of cloud condensation nuclei promotes and accelerates the Stratus-to-Cumulus Transition (SCT) due to an increase in evaporation and entrainment at the stratus top and deeper penetrating cumuli within the stratus that lead to a dissolution of the surrounding stratus via entrainment and subsequent subsidence of cold air. Furthermore, the study indicates a domain-wide reduction of clouds with moderate precipitation but a localized precipitation intensification via the convective-cloud invigoration mechanism. The interaction between AIE, the land surface characteristics and tropical sea breeze convection over Cameroon was analyzed by Grant and van den Heever (2014) for boreal summer conditions. The study reveals a weakening of the sea breeze front with increasing aerosol, due to a reduction in surface shortwave radiation and therefore surface heating, linked with less precipitation.



Stevens and Feingold (2009) and Fan et al. (2016) emphasize the need to analyze AIE cloud-regime dependent with fine-scale models to explicitly resolve the interacting processes rather than using global models with parameterizations. This is supported by the study of Marsham et al. (2013), which reveals that the West African Monsoon (WAM) representation in the UK Met Office Unified Model shows fundamental differences between realizations with explicit and parameterized moist convection.

A comprehensive overview of the current state of research on the AIE is presented in Fan et al. (2016).

This study focuses on the assessment of the aerosol impact on clouds and the atmospheric dynamics over SWA using a two-day process study. The following research questions encompass the focus of this study: What are the dominating aerosol impacts on meteorological characteristics over SWA and which spatial and temporal scales do they exhibit? Do we see changes in radiation and precipitation? To which extent does altered cloud radiative properties play a role?

This study is structured as follows: Section 2 decribes the model system COSMO-ART employed in this study together with the observational data used for evaluation. In Section 3 the *Atlantic Inflow* (AI) and *Stratus-to-Cumulus Transition* (SCT) as prevailing meteorological characteristics in SWA are introduced. The results comprise an evaluation of the modeled cloud properties with aircraft observations (Sect. 4), the COSMO-ART representation of AI (Sect. 5) and the aerosol impact hereon (Sect. 6). The study concludes with a summary and evaluation of the findings (Sect. 7).

## 2 Methods and data

### 2.1 Model framework and setup

For this study, the regional-scale model framework COSMO-ART (Consortium for Small-scale Modeling - Aerosols and Reactive Trace gases, Vogel et al., 2009) is used. COSMO-ART is based on the operational weather forecast model COSMO (Baldauf et al., 2011) of the German Weather Service (DWD). The ART extensions allow for an online treatment of the aerosol

dynamics and atmospheric chemistry. The model application of this study is accompanied with significant further developments of the emission parameterizations regarding mineral dust (Rieger et al., 2017) and BVOCs (Weimer et al., 2017). Furthermore, a parameterization for trace gas emissions from gas flaring of the oil industry was developed to reproduce the specific pollution conditions of the research area (Deetz and Vogel, 2017). The model domain comprises Ivory Coast, Ghana, Togo, Benin and the Gulf of Guinea (red rectangles in Fig. 1). The model setup is summarized in Appendix A.

The simulations using the setup denoted in Table 2 are the result of a nesting into a 5 km COSMO-ART simulation (blue rectangle in Fig. 1a) using the ICON operational forecasts (approximately 13 km grid mesh size) as meteorological boundary conditions. These cover the time period 25 June to 3 July to allow for an aerosol-chemistry spin up. The meteorological state is initialized every day at 0 UTC.

To assess the sensitivity of the ADE and AIE on the meteorological conditions, two factors $F_{ADE}$ and $F_{AIE}$ were introduced

in COSMO-ART, which allow to scale the total aerosol mass and number densities, respectively, by simultaneously preserving the underlying aerosol distribution. All aerosol modes are changed uniformly by the factors but the scaling is limited to the derivation of the aerosol optical properties in case of ADE and the aerosol activation in case of AIE. Within a simulation the constraint $F_{ADE}=F_{AIE}$ is used to allow for physically consistent results. Table 1 summarizes the realizations used in this study.





$F_{ADE}=F_{AIE}$=1.0 is used as the reference case whereas the factor variations 0.1, 0.25, 0.5, 2.0 and 4.0 are applied to assess the aerosol sensitivity. The terms *clean*, *reference* and *polluted* should be seen in a relative sense as a part of this experimental setup. They do not imply general evaluation of the SWA aerosol conditions. The period 2–3 July was selected due to the intense and persistent NLLS at Savè supersite during that time (Kalthoff et al., 2017). Furthermore, 3 July is the center of the monsoon

*Post-onset phase* and it is expected that the undisturbed monsoon condition favor and support the process studies. Since the meteorological conditions show less variation from day to day, it is assumed that, even with a focus on a very short time period, insight can be achieved that can be generalized at least qualitatively to the length of the *Post-onset phase* (22 June – 20 July).

## 2.2  Observational data

Within the DACCIWA project, an extensive field campaign took place in June–July 2016 in SWA (Fig. 1b) (Flamant et al.,

2018). The time period was selected to capture the onset of the WAM and a period characterized by increased cloudiness. The DACCIWA ground-based measurement campaign encompassed the time period from 13 June to 31 July 2016, including the three supersites Kumasi (Ghana), Savè (Benin) and Ile-Ife (Nigeria) (red dots in Fig. 1b). A complete overview of the DACCIWA ground-based measurement campaign, their supersites, instrumentation and a first insight into the available data is presented in Kalthoff et al. (2017). The DACCIWA airborne measurement campaign captured the time period from 27 June to

17 July 2016 (Flamant et al., 2018). For this study, observations of the liquid cloud properties from the CDP-100 (Cloud droplet probe, data revision 3) of the British Antarctic Survey (BAS) Twin Otter aircraft on 3 July 2016 are used for a comparison with COSMO-ART. The CDP-100 is a wing mounted canister instrument including a forward-scatter optical system to measure the cloud droplet spectrum between 2-50 $\mu$m with a frequency of 1 Hz.

## 3  SWA meteorological characteristics

Knippertz et al. (2017) separated the DACCIWA measurement campaign period into phases of similar meteorological conditions by using the precipitation difference between the coastal zone and the Soudanian-Sahelian zone. The monsoon *Onset phase* is identified as the period 16–26 June. The monsoon *Post-onset phase*, characterized by undisturbed monsoon conditions, is identified between 22 June and 20 July. Especially in the *Post-onset phase*, SWA is frequently covered by NLLS (Knippertz et al., 2017; Kalthoff et al., 2017). The formation mechanisms of NLLS are not entirely clear. Figure 2 is a schematic to em-

phasize the general meteorological patterns relevant for the subsequent process study. In the nighttime a frequent occurrence of a nocturnal low-level jet (NLLJ, black arrows in Fig. 2a) can be observed with a jet maximum at 300 m above ground level (AGL) of around 6 m s$^{-1}$ (Kalthoff et al., 2017). The NLLS is formed at the height of the jet maximum via shear-driven vertical mixing of moisture and maintained via cloud-top radiative cooling and cold advection (Schuster et al., 2013). Also topographic lifting (Schuster et al., 2013; Adler et al., 2017) as well as vertical cold air advection in gravity waves and cloud

formation upstream of existing clouds contribute to the NLLS formation (Adler et al., 2017). After sunrise a gradual SCT takes place (e.g. Kalthoff et al., 2017). This is accompanied by a lifting of the cloud base (Fig. 2b). The increase of the liquid water path shortly after sunrise is related to the growth of the convective boundary layer. Further analysis of this topic will be




conducted within the framework of DACCIWA using observational data gathered during the ground-based field campaign. In the morning hours the maximum spatial coverage of NLLS can be observed (Fig. 2b). In the subsequent hours the NLLS deck breaks up to cumuliform clouds (Fig. 2c). Adler et al. (2017) identify a regular occurrence of a stationary coast-parallel front over SWA about 30 km inland that propagates northwards during undisturbed monsoon conditions after about 16 UTC. Similar characteristics were described in Grams et al. (2010) for Mauritania. Grams et al. (2010) indicate that the stationarity results from a balance between horizontal advection with the monsoon flow over the ocean and inland turbulence in the boundary layer that mixes the momentum vertically (Fig. 2d). With the reduction in turbulence in the afternoon the front begins propagating inland. The studies of Grams et al. (2010) and Adler et al. (2017), both based on modeling studies using the COSMO model, highlight the need to distinguish this feature from the land-sea breeze and the sea breeze front, since the dominating monsoon flow suppresses the formation of a land wind during night. In the following, we use the term *Atlantic Inflow* (AI) as proposed by Grams et al. (2010), which is connected with an AI front and an AI airmass located behind the front. The potential feedbacks between the aerosol on the one hand and the NLLS and the AI front on the other hand have not been quantified for SWA.

## 4 Evaluation of modeled cloud properties with aircraft observations

To evaluate the modeled cloud properties, observations of the research aircraft *British Antarctic Survey (BAS) Twin Otter* on 3 July 2016 between 10:47 UTC and 14:06 UTC (flight number TO-02) are used, capturing the Lomé-Savè area. The following figures show the flight path and altitude (Fig. 3) as well as the observed and modeled cloud droplet number concentration (CDNC, Fig. 4a,b) and effective radii (Fig. 4c,d). The aircraft position between 10:45–11:30 UTC, 11:30–12:30 UTC, 12:30–13:30 UTC and 13:30–14:06 UTC is shown in blue, grey, red and black, respectively, for the flight track (Fig. 3a) and the altitude (Fig. 3b). For a more robust statistical comparison of the observed and modeled cloud location, the comparison with COSMO-ART is not realized along the flight track but by using the cubes that are spanned horizontally by the rectangles around the flight track sections for 11-14 UTC (according to the hourly output of COSMO-ART) and vertically by the lowest 2.3 km AGL in accordance to the Twin Otter maximum flight altitude during this flight. The observed and modeled CDNC and effective radii are compared via boxplots (Fig. 4) for the flight track sections at 2 July between 11 UTC and 14 UTC. The boxplot colors follow the definition in Figure 3. For 11 UTC, the observations are omitted since the Twin Otter did not penetrate clouds during that time. The modeled CDNC (Fig. 4b) are generally higher than the observed ones (Fig. 4b) but both stay below a median of 400 cm$^{-3}$. The model shows a general trend of increasing median CDNC with time. This is expected during the SCT, since cumulus clouds tend to have a higher CDNC than stratus. Miles et al. (2000) provided a data base of observed cloud properties of low-level stratiform clouds. For example for the Madeira Islands they identified CDNC around 50 cm$^{-3}$ for nocturnal stratiform clouds and around 300 cm$^{-3}$ for cumulus and stratocumulus. The smaller CDNC at 14 UTC is likely related to a reduced number of observations in clouds due to the approach of the Twin Otter at Lomé. In addition to the uncertainty in the modeled aerosol number and number of activated particles also the limited number of cloud penetrations of the Twin Otter can contribute to the deviations. The Twin Otter did not fly continuously in clouds but performed descents





and ascents (see Fig. 3b). The modeled increase in CDNC with time in Figure 4b is related to a slight decrease in the effective radii in Figure 4d. Generally, the observed and modeled median effective radii are around 6 $\mu$m and thus in good agreement.

## 5 Model representation of the Atlantic Inflow (AI)

All the realizations in Table 1 exhibit the AI phenomenon. Following Grams et al. (2010), the AI front position can be estimated
by the location at which a specific isentrope of virtual potential temperature $\theta_{v,s}$ crosses a specific height $h_s$. For Mauritania Grams et al. (2010) used $\theta_{v,s}$=310 K at the surface pressure level. For this study, reasonable results are achieved by using the potential temperature $\theta_s$=302 K and the height $h_s$=250 m AGL. These values are selected empirically and are related to the COSMO-ART results of this study. They do not claim general applicability. However, in contrast to the definition in Grams et al. (2010), here it seems more appropriate to use a level elevated from the ground to identify the front, since the frontal
gradients are most prominent at the height of the NLLJ axis (about 250 m AGL), whereas the frontal passage is hard to detect in surface observations (N. Kalthoff, personal communication). Figure 5 shows the location of the AI front between 15 and 22 UTC for 2 July 2016 (Fig. 5a) and 3 July 2016 (Fig. 5b) in the reference case. Although the focus is on 2 July, 3 July is added to underline that the AI is a robust feature occurring frequently over SWA, which is also indicated by the results of Adler et al. (2017). The $\theta_s$ method for the AI front location is only an estimation, since the potential temperature is also altered
by surface conditions and diabatic effects. We focus on the front location of the time period 15–22 UTC that coincides well with the wind speed patterns as presented subsequently. With the increasing nighttime cooling over land after 22 UTC, the temperature gradient between the AI post-frontal and pre-frontal airmass diminishes, impeding the localization of the front. Figure 5 shows an inland propagation of the AI front with time (coded by the line colors). Generally the front is parallel to the coast. This is most obvious for the domain west of 2°W. In contrast, the Lake Volta area and also the area east of the Atakora
Mountains show higher variability in the frontal location. Lake Volta is a flat area with fixed surface temperatures in the model and reduced roughness, likely affecting the frontal propagation. For the following analysis, the focus is set to Ivory Coast (7.5–3.0°W). The distance between the hourly frontal locations reveals that in the evening (approximately 15–18 UTC) the propagation velocity is slow at the beginning but then increases. At 15 UTC the front is located about 100 km inland. Before 15 UTC the AI front is not detectable, since the inland area is subject to warming, which shifts $\theta_s$ in coastal direction. However,
between 11 UTC and 15 UTC already a horizontal wind speed gradient develops in an area between the coastline and 100 km inland with enhanced (reduced) values over the Gulf of Guinea (over land). Meridional vertical transects of wind speed and potential temperature for this time period are provided in Appendix B. Interestingly, these transects also emphasize the reduced monsoon flow further inland with the development of the AI front (compare 6–7 °N between Fig. 17a and Fig. 17b-e), which is also shown schematically in Fig. 2d. The estimated frontal propagation velocity for the reference case on 2 July stagnates
around 7 m s$^{-1}$ after 19 UTC. This is on the same order of magnitude as the findings of Grams et al. (2010) of 10±1 m s$^{-1}$ for Mauritania. To gain further insight in the general structure of the AI, Figure 6 shows the meridional vertical transects along 5.75°W (central Ivory Coast) for the reference case. Figure 6a shows the horizontal wind speed as shading and the isentropes of 301, 302 and 303 K as solid black contours. As described above, the AI front (vertical dashed line) is identified



by using the 302 K isentrope (bold solid line). Several general characteristics can be concluded from Figure 6a: (1) The AI front marks the location of strongest horizontal gradients in wind speed and potential temperature. (2) The post-frontal wind speeds are significantly higher than the pre-frontal wind speeds. The post-frontal area reveals a band of high wind speeds below approximately 900 m ASL with a maximum at around 300 m AGL. This is typical of the NLLJ with the jet axis highlighted by

the horizontal dashed line. The entire post-frontal area is affected by this low-level wind band or "blanket" when considering the entire SWA domain. (3) The pre-frontal wind speed is vertically more homogeneous than in the post-frontal area indicating that the AI front is also a border between a predominant well-mixed boundary layer pre-frontally and ongoing stabilization post-frontally. (4) The post-frontal airmass is characterized by cooler temperatures than the pre-frontal area. Therefore the AI frontal passage is related to an increase in wind speed and a decrease in temperature. (5) In agreement to the findings of Grams

et al. (2010) the flow patterns are structurally similar to that of a density current where fast moving cold air and surface friction lead to the formation of an overhanging nose and a head that can extend to higher altitudes than the tail (Simpson, 1969; Sun et al., 2002). Vertical extensions of the head of about 1 km are found for atmospheric density currents (Simpson, 1969), which agrees with the flow in Figure 6a. Sun et al. (2002) emphasize that the wind surge behind the nose, that propagates close to the ground, leads to strong turbulent mixing. This can also be observed in this process study when focusing on the vertical transect

of TKE (Fig. 6b). Generally, the post-frontal area shows higher TKE values than pre-frontally. Especially in the area behind the nose TKE is enhanced. Strongest turbulence is not within the jet axis (horizontal dashed line) but below (near the surface) and above due to shear. The location of the 302 K isentrope, which is used for the AI front detection, corresponds well with the layer of increased TKE at the upper border of the AI. It is expected that the near-surface turbulence favors the vertical mixing of moisture as indicated e.g. by Schuster et al. (2013).

The study of Adler et al. (2017) reveals that the AI frequently occurs under undisturbed monsoon conditions over SWA, reaching Savè around 21 UTC. This agrees well with the latitudinal AI front evolution in this study (not shown).

## 6 Aerosol impacts and mechanisms

### 6.1 First insight in the aerosol impact on AI

After describing the general AI properties in Section 5, this section presents first insight into the aerosol influence on AI. Figure

7 shows the horizontal wind speed difference at 250 m AGL ($h_s$) on 2 July 22 UTC between the clean and the reference case together with the corresponding AI front locations. The wind speed difference exhibits a filament structure in zonal direction that covers nearly the entire SWA domain. Furthermore, it propagates inland with time (not shown). Especially over Ivory Coast a coherent pattern can be observed with a spatial shift between the two AI fronts with that of the clean case (black dashed line in Fig. 7) ahead of the reference case front (black solid line in Fig. 7). This anomaly pattern results from the fact that the post-

frontal wind speeds are generally higher than the pre-frontal wind speed, as shown in Fig. 6a. To assess the aerosol impact on the vertical structure of the AI, Figure 8 shows the meridional vertical transect of wind speed and potential temperature for the clean (Fig. 8a) and polluted case (Fig. 8b) in the same way as presented for the reference case in Figure 6a. When comparing the results between the clean and polluted cases, and by considering the reference case (Fig. 6a) as intermediate, aerosol-specific



characteristics can be identified in addition to the general AI characteristics presented in Section 5. Whereas the temperature characteristics over the ocean are similar for the realizations, the inland temperature decreases with increasing aerosol amount. This is especially visible in the pre-frontal area (Fig. 8). In the polluted case the advective cooling is more effective, since the daytime inland near-surface air is a priori cooler due to a lower sensible heat flux from aerosol extinction. The reduced

ocean-land temperature gradient in the polluted case leads to reduced temperature contrasts at the AI front (compare the 302 K isentrope for the clean case (bold line in Fig. 8a) and the polluted case (bold line in Fig. 8b)). With the change in the ocean-land temperature gradient, the AI frontal position and the NLLJ strength and vertical extension is altered. The higher the aerosol amount, the more the AI front is lagging behind and the weaker the NLLJ. In the polluted case the vertical extension of the inland NLLJ and its wind speed in the jet axis is reduced by about 150 m and 2–3 m s$^{-1}$, respectively. The AI frontal difference

averaged over Ivory Coast at 21 UTC is 10 km between the clean and reference case and 20 km between the clean and the polluted case. With the decrease in temperature with increasing aerosol amount, the pre-frontal wind speed generally increases (contrarily to the post-frontal area). This leads, with respect to the polluted case (Fig. 8b), to some areas of increased wind speed in the pre-frontal area, at a height that is typical of the NLLJ. Generally, the polluted case is characterized by a blurring of the pre- and post-frontal temperature and wind speed differences.

The temporal evolution of the median coastal distance of the AI front over Ivory Coast is presented in Figure 9 for the six realizations. The polluted case (solid red) denotes a special case in Figure 9. The other realizations with scaling factors between 0.1 and 2.0 show a systematic behavior. As expected, the front propagates inland with time. The higher the aerosol amount (dashed blue to solid green) the slower the inland propagation. This leads to a median spatial difference of about 27 km between factor 0.1 (dashed blue) and 2.0 (solid green) at 22 UTC. Furthermore, Figure 9 reveals two regimes, one before and one after

17 UTC. For the latter the frontal propagation diverges according to the aerosol amount as described above. Before 17 UTC an opposite behavior can be observed leading to the circumstances that with less aerosol the AI front is closer to the coastline. Therefore the starting point for the inland propagation at 15 UTC is not equal for all realizations but a reversed order can be observed compared to the situation at 22 UTC. The underlying mechanisms for the occurrence of these two regimes that switch around 17 UTC is assessed in the subsequent section. With a further increase in the aerosol amount, as realized in the polluted

case (solid red line in Fig. 9), the ocean-land temperature gradient is reduced as shown in Figure 8b. The AI front evolution is therefore less pronounced than for the other realizations. In the eastern part of Ivory Coast the location of $\theta_s$ persists inland and does not form a coherent front near the coast. Averaging the frontal location over Ivory Coast therefore leads to a temporal evolution, which does not follow the behavior of the other realizations. By reducing the benchmark of $\theta_s$ from 302 K to 301 K, also the polluted case follows the trend of a weaker frontal propagation with increasing aerosol (grey line in Fig. 9). In this

polluted case, with its cooler lower layers, the 301 K isentrope better represents the frontal location, as also visible in Figure 8b.

## 6.2 Aerosol-AI impact mechanism

After diagnosing the characteristics of AI and the AI changes with changing aerosol amounts in Section 5 and 6.1, respectively, the question of the underlying feedback mechanism arises. The stationarity of the AI front near the coast in the early afternoon





is related to the balance between the onshore directed monsoon flow over the ocean and the turbulence over land (e.g. Grams et al., 2010). Therefore the change in the turbulence can alter the balance and lead to differences in the AI front propagation (turbulence mechanism). With increased aerosol amounts, a near-surface cooling (relative to the realizations with less aerosol) is expected during daytime (as observed in Fig. 8b), either due to the ADE via scattering and absorption on aerosol particles or

due to to AIE via an increased albedo with reduced cloud droplet effective radii (Twomey effect). A reduced surface heating with increased aerosol amounts might lead to an increase in surface pressure. By considering the fact that the Sea-Surface Temperature (SST) is fixed in COSMO-ART and that the surface temperature over the ocean will therefore not be subject to substantial changes, a reduction in the land-sea pressure gradient can be expected, which could affect the AI front propagation (pressure gradient mechanism). In order to further elaborate this, Figure 10 shows the spatial distribution of total cloud water

(Fig. 10a) and precipitation (Fig. 10b) for the reference case on 2 July, 15 UTC. The red line denotes the AI front. Clouds and precipitation occur primarily in the AI post-frontal area over Ivory Coast due to convergence and vertical lifting, upstream of mountain areas due to topographic lifting (especially at the Mampong Range and the Atakora Mountains) and via localized convection (primarily in the AI pre-frontal area over Ivory Coast). To shed light on potential effects from the turbulence mechanism and the pressure gradient mechanism, Figure 11 shows the differences in surface net downward radiation (Fig.

11a), 2-m temperature (Fig. 11b) and sea level pressure (Fig. 11c) between the reference and the clean case. In addition, Figures 11b,d,f present the same variables but for the areas that are cloud free in both realizations to exclude effects from displaced clouds and to highlight the ADE in a cloud-free environment.

The differences in surface meteorological quantities presented in Figure 11 reveal a clear signal. The following values in brackets indicate the median and the $99^{th}$/$1^{th}$ percentile of the surface quantities considering the cloud-free inland area. With

increasing aerosol, more downward shortwave radiation is scattered and absorbed, leading to an average decrease in surface net downward shortwave radiation (-37 W m$^{-2}$, -185 W m$^{-2}$; Fig. 11b). The decrease in incoming shortwave radiation leads to a decrease in 2-m temperature (-0.5 K,-2.5 K; Fig. 11d). The temperature decrease furthermore leads to a domain-wide inland surface pressure increase (+0.16 hPa, +0.45 hPa; Fig. 11f). Omitting the negative pressure anomaly over lake Volta due to the fixed SST, slightly higher domain-wide pressure anomalies are found (+0.17 hPa, +0.45 hPa; Fig. 11f).

To prove the hypothesis that the surface pressure difference is caused by the temperature difference, the surface pressure over Ivory Coast at 15 UTC (over land and in cloud-free areas) is estimated from the pressure and temperature at 850 hPa using the barometric equation. This approach yields a spatially averaged value of +0.12 hPa that broadly agrees with the modeled value of +0.17 hPa (compare Fig. 11f). It can be concluded that the pressure changes are dominated by changes in low-level temperature. In the cloudy areas around the SWA mountains a higher pressure difference can be observed (Fig. 11e), indicating

that also cloud-radiative effects are contributing. To assess whether the reduction of incoming shortwave radiation due to clouds is related to a change in the cloud water content and therefore the optical thickness of the clouds or due to the Twomey effect with a change in cloud droplet number concentration (CDNC) and effective radius, Figure 12 exhibit the Empirical Cumulative Distribution Function (ECDF) with respect to the COSMO-ART realizations of the CDNC (Fig. 12a), cloud droplet effective radius (Fig. 12b), cloud water (Fig. 12c) and precipitation (Fig. 12d). This figure corresponds to the cloud and precipitation

patterns presented in Figure 10 for 2 July, 15 UTC. A strong susceptibility of the CDNC and effective radii towards a change





in the aerosol amount can be observed (Fig. 12). The factor variation from 0.1 to 4.0 leads to an increase in the median CDNC by one order of magnitude from 100 to 1000 cm$^{-3}$ (Fig. 12a) and a reduction in the median effective radius from 9 to about 3.5 $\mu$m. When considering the green and red curves in Figure 12, which are related to an aerosol change symetrically around the reference case (black), the effect on the CDNC and effective radius is nonlinear (e.g. Bréon et al., 2002). An aerosol increase
(solid green and red lines) has significantly stronger impacts than the aerosol decrease (dashed green and red lines).

In contrast to these remarkable changes, the effect on cloud water and precipitation (Fig. 12c,d, respectively) is insignificant. Except of the polluted case (solid red lines) all realizations show similar ECDFs, indicating that the aerosol increase neither leads to a cloud water increase due to precipitation suppression or due to enhanced water vapor condensation on the aerosol particles nor a cloud water decrease via enhanced evaporation. The polluted case shows a tendency of precipitation decrease
(increase) for the weak (strong) precipitating areas, related to an increase (a decrease) in cloud water. This effect of greater local rainfall amounts is in agreement with the findings of Saleeby et al. (2014) likely via the convective-cloud invigoration mechanism. However, the deviations from the other realizations are small. Figure 12 reveals that the aerosol impact on radiation via the Twomey effect is very likely dominating the cloud-radiation interaction, whereas the cloud optical thickness impact (via a change in the amount of cloud water) is of minor importance. The weak precipitation response to the changing aerosol amount
underlines the finding that the radiation and its variation is the key player in the observed changes over SWA due to the ADE in and outside of clouds and the Twomey effect. There is ongoing work within DACCIWA with respect to Large Eddy Simulations (LES) of aerosol-atmosphere interactions. It will be of interest to see whether the COSMO-ART results are consistent with the outcomes of these studies on smaller scales.

The turbulence and pressure gradient mechanisms are counteracting. With respect to the turbulence mechanism, a reduced
heating weakens the turbulence in the PBL. Therefore the AI balance between the monsoon flow and the inland turbulence is shifted to the monsoon flow, favoring an inland propagation. Regarding the pressure gradient, a reduced heating decreases the land-sea pressure gradient, shifting the AI balance in the opposite direction and suppressing the inland propagation. When going back to the temporal evolution of the coastal distance of the AI front in Figure 9, both mechanisms are evident and cause the two observed regimes before and after 17 UTC. The first regime includes the stationary phase of the AI front near the coast.
With the decrease in incoming solar radiation with increasing aerosol the turbulence decreases and therefore the stationary front location shifts inland. Unfortunately, the AI front detection via the $\theta_s$ method fails for the time period earlier than 15 UTC. Therefore the total difference in the stationary AI front location with changing aerosol cannot be assessed. Nevertheless, it is interesting that the location of the AI front during its stationary phase over Ivory Coast could be used as a proxy for the aerosol burden in that area (under otherwise identical conditions).

For the time period after 17 UTC, when turbulence has decreased sufficiently, the pressure gradient mechanism dominates, because the AI front in the clean case - although lagging behind at 15 UTC, is 11 km ahead of the reference case at 22 UTC (Fig. 9).

Figure 13 summarizes the counteracting components turbulence and pressure difference that govern the inland propagation of the AI front by comparing the temporal evolution of the differences between the reference and clean case (dashed lines) and the
35 polluted and reference case differences (solid lines) in surface sensible heat flux (red, positive downward) and surface pressure



(blue), spatially averaged for the AI pre-frontal area over Ivory Coast. The temporal evolution clearly shows that the sensible heat flux differences (and the absolute values itself) decrease strongly with time in contrast to the pressure differences. After sunset (e.g. 18:24 UTC at Kumasi) the sensible heat flux is negligible but the pressure differences continue. In fact, the altered land-sea pressure gradient is maintained till the AI front and the subsequent cool airmass have passed the area and compensates

the differences (not shown). It is expected that the high moisture in the monsoon layer prevents it to cool significantly and to reduce the differences that developed during daytime. The factor increase of 4 from 1.0 to 4.0 reduces (increases) the sensible heat flux (sea level pressure) more than the increase from 0.25 to 1.0, in agreement with the findings of the sensitivity of CDNC and effective radius in Figure 12.

The monsoon flow over SWA is driven by the temperature gradient between the cool SSTs over the eastern equatorial Atlantic
Ocean that are fixed in the model and the Saharan Heat Low that is not part of the modeling domain. With this location of the modeling domain, changes in the aerosol amount can serve as an amplifier for the monsoon flow that is able to increase or decrease the temperature gradients and thereby the AI front characteristics. In agreement, Grant and van den Heever (2014) show that the sea breeze front over Cameroon weakens with enhanced aerosol number concentration. Longwave cooling is not significantly reduced, likely due to the water vapor saturation in the monsoon layer (not shown). In contrast, the coherent
differences in 2-m temperature and pressure, which were observed at 15 UTC (Fig. 11), also persist during nighttime. The daytime heating of the land, stronger in the clean case and weaker in the polluted case, persists during night and exceeds potential effects from longwave cooling. The differences between the realizations are finally equalized by the passage of the AI front and post-frontal airmass.

### 6.3 Aerosol-SCT impact mechanism

In addition to the aerosol impact on AI, also impacts on the SCT can be observed. Figure 14 shows the vertical transect of modeled cloud water between Lomé and Savè (Fig. 1b) regarding the clean case (left) and the reference case (right) for 2 July 10 UTC (top) and 11 UTC (bottom). The red shaded area below the cloud layer denotes the development of the convective boundary layer (CBL) identified by $d\theta/dz<0$. The black (red) solid line shows the top of this unstable layer regarding the reference (polluted) case to allow for comparison between the three realizations. After sunrise the CBL starts to evolve. Via the
same mechanism as described in Section 6.2, less shortwave radiation reaches the ground with increased amounts of aerosol and therefore also the surface sensible heat fluxes decrease. This leads to a decelerated daytime CBL development and with that to a reduction of the cloud base height (Fig. 14, left). To underline that this effect is visible not only in the Lomé-Savè transect but for the entire SWA region, Figure 15 shows the temporal evolution of the spatial average of total cloud cover (Fig. 15a), total cloud water (Fig. 15b) and the cloud base height (Fig. 15c) over SWA, for the clean (blue dashed), reference
(black solid) and polluted case (red solid). Between 21 UTC and the time of sunrise (5:30 UTC) the cloud cover increases (Fig. 15a) due to clouds that are advected onshore or develop inland. This is linked with a reduction in the mean cloud base (Fig. 15c). Between 1 UTC and 7 UTC the clean case shows lower cloud base values than the reference and polluted cases. A detailed analysis reveals that this deviation is not related to NLLS but to mid-level clouds over the Lake Volta Basin and in the northwestern part of the domain (not shown). After sunrise it is assumed that the NLLS intensifies via vertical mixing



of moisture in the developing convective PBL. With respect to the spatial average in Figure 15c this leads to a reduction in mean cloud base height. The maximum cloud cover (Fig. 15a) is related to the minimum cloud base (Fig. 15c), underlining the dominance of NLLS. After reaching the cloud cover maximum, the SCT continues, which is related to a lifting of the cloud base and a decrease in cloud cover. For this SCT a clear temporal shift of about one hour can be observed between the clean

and the reference case and two hours between the clean and the polluted case. The realizations with increased aerosol amounts react slower to the insolation after sunrise, reach the NLLS maximum coverage later and start later with the SCT as observed for the Lomé-Savè transect in Figure 14. After 15 UTC this finally leads to a cloud cover that is increased compared to the clean case (Fig. 15a) implying an additional reduction in surface shortwave radiation that can be used for further cooling the surface and decelerating the AI front. The cloud water (Fig. 15b) shows a similar temporal shift with increasing aerosol amounts as

for the cloud cover and cloud base. The weakening of the SCT with a higher aerosol burden leads to reduced amounts of cloud water after 13 UTC (Fig. 15b) likely due to reduced convective activity. However, during nighttime, the polluted case uniformly shows higher cloud water values than the clean and reference cases.

Figure 18 in Appendix C shows the cloud analysis by restricting to the clouds below 1500 m AGL to assess the sensitivity of the spatial averaging towards the considered vertical column. The cloud cover (Fig. 18a) shows a similar temporal evolution as

presented in Figure 15a. The cloud water and cloud base temporal evolution in the lowest 1500 m AGL (Fig. 18b,c) show less variations between the realizations compared to Figures 15b,c. However, the temporal shift in the onset of the SCT is obvious in both figures. As expected, the initiation of the cloud base increase via the SCT occurs earlier when considering only the clouds below 1500 m AGL in the averaging (compare Fig. 18c with Fig. 15c).

The aerosol feedback process study simulations presented in Section 6.2 and Section 6.3 revealed several mechanisms relevant

for SWA, affecting the location and propagation of the AI front and the temporal evolution of the SCT. In the following section a proposal for a conceptual model will be presented.

### 6.4 Conceptual model of aerosol-atmosphere interactions in SWA

This section aims to synthesize the findings that have been obtained with this aerosol feedback process study. We showed that AI affects the entire SWA domain through the course of the day via cold air advection, the NLLJ that can be found in the AI

post-frontal area and convergence-induced convection and precipitation. Two distinct meteorological responses to changes in the amount of aerosol via ADE and the Twomey effect were identified: *1. A spatial shift of the Atlantic Inflow (AI)* and *2. a temporal shift of the Stratus-to-Cumulus Transition (SCT)*.

Figure 16 shows a conceptual scheme that combines both responses. The bigger loop is related to the first response (AI) and the smaller loop to the second (SCT). Following the AI loop in Figure 16, the increase in the amount of aerosol (number and

mass) by a factor of 4 (0.25 to 1.0) is the initial perturbation of the system. The subsequent numbers in parenthesis are related to the median value over Ivory Coast (cloud-free inland areas) on 2 July 15 UTC to provide guiding values for the denoted changes.

Via ADE the aerosol increase leads to a decrease in surface net downward shortwave radiation (-37 W m$^{-2}$) and surface temperature (-0.5 K). Previous studies showed that till the early afternoon, the AI front is stationary near the coast due to the



balance between the monsoon flow from the sea and the sensible heat flux (turbulence) over land. With the afternoon decrease in sensible heat flux, the AI front propagates inland. This study showed that the decreased surface heating leads to a positive pressure anomaly over land (+0.16 hPa) and with that to a reduced land-sea pressure gradient. The latter is more persistent than the sensible heat flux that vanishes around sunset (compare Fig. 13). The reduced pressure gradient leads to a reduced

AI frontal velocity and therefore to a southward shift in the case of increased aerosol (11 km on 2 July 22 UTC). The post-frontal area is characterized by stronger wind speeds in the lowest 1000 m AGL with the maximum around 250 m AGL that is characteristic of the NLLJ. Therefore an AI frontal shift leads to a shift in the NLLJ inland propagation. Since the AI frontal propagation is linked to convergence-induced convection and convective precipitation, also a meridional shift of the AI-related precipitation is observed. These effects are primarily related to the afternoon but the AI frontal and NLLJ shift also leads to a

shift in the inland propagation of coastal NLLS with a similar spatial magnitude as observed for the AI front (not shown).

The AI loop denoted in Figure 16 includes a further mechanism, related to the counteracting effects of the monsoon flow over the ocean and the sensible heat flux over land in the stationary phase of the AI front. With increasing aerosol the inland sensible heat flux decreases, which relocates the front farther from the coast. Therefore with increased aerosol the AI frontal inland propagation starts farther from the coast but is slower than in the low aerosol case due to the reduced land-sea pressure

gradient as soon as the turbulence has declined after sunset.

The SCT loop is coupled to the AI loop via the decrease in surface shortwave radiation and temperature. This study pointed out that the deficit in surface heating due to ADE and cloud brightening via the Twomey effect lead to a decrease in sensible heat flux and therefore to a delayed development of the CBL. The lower CBL height leads to a lower cloud base and therefore to a later SCT and breakup of the closed cloud layer to scattered cumuli (compare Fig. 15a). Both loops are initialized after sunrise

with the input in shortwave radiation. The SCT loop implies a positive cloud cover anomaly after 15 UTC with increasing aerosol. Sunset is around 18:30 UTC. Although the AI front already starts penetrating inland around 14-15 UTC, approximately a 3.5-hour period is available for an additional surface cooling from the later cloud-layer breakup. This is a pathway for a further deficit in surface shortwave radiation and surface heating that could further weaken the AI loop as emphasized by the red arrow in opposite direction in Figure 16. However, the latter coupling between the two loops is only hypothesized. A future study

needs to assess the significance of the contribution in inland surface pressure increase that comes from the deficit in shortwave heating via the later cloud-layer breakup.

The mechanisms described in Figure 16 raise the question about the possibility to generalize these results. The AI feature is very likely a regular phenomenon under undisturbed monsoon conditions as confirmed by previous studies that focus on longer time periods. Within this process study the AI frontal shift was obvious for both days in the evening. However, the results presented

above are related to Ivory Coast that shows a more coherent AI frontal pattern than the eastern part of the domain, likely related to topographic features. This conceptual picture reveals radiation as a key player governing the feedbacks, either via ADE or via a change in cloud albedo (Twomey effect). The AIE assessment within the process study reveals the known mechanisms, in particular the increase (decrease) of the CDNC (effective radius) with an increase in the aerosol number concentration. However, the AI-related clouds and precipitation reveal, aside from a meridional shift, no statistically significant difference.





Although, the possibility for substantial effects from AIE cannot be excluded, a conceptual view as presented for the radiative effects has to be left for subsequent studies.

## 7    Conclusions

This study focused on southern West Africa (SWA) to assess the implications of aerosols on clouds and atmospheric dynamics using a process study with the regional model COSMO-ART on 2–3 July 2016, a time period in the well-established West African Monsoon (WAM) without impacts of Mesoscale Convective Systems. The results revealed an elongated front over SWA that develops during daytime between the monsoon flow over the ocean and the turbulence over land being stationary near the coast around noon and propagating inland in the evening. This phenomenon has been identified for several African coastal regions and was conceptually separated from the classical land-sea breeze. Based on Grams et al. (2010) we used the term *Atlantic Inflow* (AI). The AI post-frontal area is characterized by a distinct decrease in temperature and increase in wind speed and relative humidity, emphasizing that the nocturnal low-level jet (NLLJ) in SWA is a widespread phenomenon related to AI.

Changing the aerosol number and mass in COSMO-ART, the aerosol direct effect (ADE) and indirect effect (AIE) was quantified, indicating a considerable sensitivity of the AI frontal location towards changes in the aerosol amount. With increasing aerosol the AI front shows reduced propagation velocities over Ivory Coast leading to frontal displacements of 10-30 km. Grant and van den Heever (2014) modeled a similar behavior for the sea breeze over Cameroon. Longwave cooling influences the AI pre-frontal area but even after sunset the positive temperature anomaly from daytime solar heating persists and dominates. In addition to the effect on AI, the decrease in near-surface heating leads to a delayed Stratus-to-Cumulus Transition (SCT) via a later onset of the convective boundary layer. We synergized this subtle aerosol-atmosphere feedback in a new conceptual model combining the AI and SCT loops (Fig. 16). Furthermore, we hypothesize that the additional radiation deficit due to the later SCT leads to a further weakening of AI.

The results exhibit the radiation as the key player governing the aerosol affects on SWA atmospheric dynamics during boreal summer, via ADE and the Twomey effect. In contrast, effects on precipitation are small. Saleeby et al. (2014) identified AIE as relevant for the SCT over tropical oceans with an accelerated transition with increasing aerosol. This study identified ADE and the Twomey effect as predominant for the SCT over tropical land areas with a decelerated transition with increasing aerosol. The importance of ADE on monsoon-related processes has also been shown by Lau et al. (2017) for the Indian monsoon. For Northern India, they reveal that the ADE dominates large-scale aerosol-monsoon interactions. A detailed literature study suggests that in the current aerosol research, ADE and the cloud-radiation interactions are underrepresented. Especially with respect to monsoon regimes, a special focus should be set on ADE. Whether the AI frontal displacement is detectable in long-term observations is left for subsequent studies. A potential strategy is the analysis of the AI front around noon via remote sensing cloud observations from past to present by assuming a positive trend in the aerosol burden. It is expected that the daytime AI front location has shifted landwards from the past to current conditions but also other phenomena (e.g. decadal SST variations) have the potential to affect the front location.



*Data availability.* The underlying research data are available upon request from the corresponding author.

*Competing interests.* The authors declare that they have no conflict of interest.

*Special issue statement.* This article is part of the special issue *Results of the project "Dynamics–aerosol–chemistry–cloud*
5 *interactions in West Africa" (DACCIWA)*

## Appendix A: COSMO-ART model configuration

*Tab. 2*

## Appendix B: Early AI evolution on 2 July 2016

*Fig. 17*

10 ## Appendix C: Temporal evolution of clouds below 1500 m AGL

*Fig. 18*

*Acknowledgements.* The research leading to these results has received funding from the European Union 7th Framework Programme
(FP7/2007-2013) under Grant Agreement no. 603502 (EU project DACCIWA: Dynamics-aerosol-chemistry-cloud interactions in West
Africa). We thank the German Weather Service (DWD) for providing access to the ICON forecast data.





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





**Table 1.** Overview of the COSMO-ART realizations capturing the variation in the aerosol amount with respect to the Aerosol Direct Effect (ADE) and Aerosol Indirect Effect (AIE).

| Abbreviation | Description of Simulation |
|---|---|
| $AIE_{0.1}ADE_{0.1}$ | $F_{AIE} = 0.1$ and $F_{ADE} = 0.1$ |
| $AIE_{0.25}ADE_{0.25}$ | $F_{AIE} = 0.25$ and $F_{ADE} = 0.25$ (clean case) |
| $AIE_{0.5}ADE_{0.5}$ | $F_{AIE} = 0.5$ and $F_{ADE} = 0.5$ |
| $AIE_{1.0}ADE_{1.0}$ | $F_{AIE} = 1.0$ and $F_{ADE} = 1.0$ (reference case) |
| $AIE_{2.0}ADE_{2.0}$ | $F_{AIE} = 2.0$ and $F_{ADE} = 2.0$ |
| $AIE_{4.0}ADE_{4.0}$ | $F_{AIE} = 4.0$ and $F_{ADE} = 4.0$ (polluted case) |





**Table 2.** COSMO-ART model configuration used for this study.

| Characteristics | Description |
| --- | --- |
| Model version | COSMO5.1-ART3.1 |
| Time period | 2–3 July 2016 |
| Simulation domain | 9.0°W–4.4°E, 3.0°N–10.8°N |
| Grid mesh size | 2.5 km (0.0223°) |
| Vertical levels | 80 up to 30 km (28 in the lowest 1.5 km ASL) |
| Meteorological boundary and initial data | COSMO-ART (5 km grid mesh size using ICON operational forecasts from DWD) |
| Pollutant boundary and initial data | COSMO-ART (5 km grid mesh size using MOZART, 2017) GlobCover (2009) land use data CCSM (2015) plant functional types |
| Cloud microphysics | Two-moment microphysics scheme (Seifert and Beheng, 2006) |
| Pollutant emissions | Mineral dust (online): Rieger et al. (2017) using HWSD (2012) Sea salt (online): Lundgren et al. (2013) DMS (online): using Lana et al. (2011) BVOCs (online): Weimer et al. (2017) Biomass burning (prescribed/online): Walter et al. (2016) using GFAS (CAMS, 2017) Anthropogenic (prescribed): EDGAR (2010) Gas flaring (prescribed): Deetz and Vogel (2017) |
| Aerosol dynamics | MADEsoot (Riemer et al., 2003; Vogel et al., 2009) Secondary inorganic aerosol: ISORROPIA II (Fountoukis and Nenes, 2007) Secondary organic aerosol: VBS (Athanasopoulou et al., 2013) |
| Chemical mechanisms | Gas phase chemistry: RADMKA (Vogel et al., 2009) |
| Aerosol direct effect (ADE) | Vogel et al. (2009) |
| Aerosol indirect effect (AIE) | Warm phase: Bangert (2012) and Fountoukis and Nenes (2005) Cold phase: Philipps et al. (2008) |



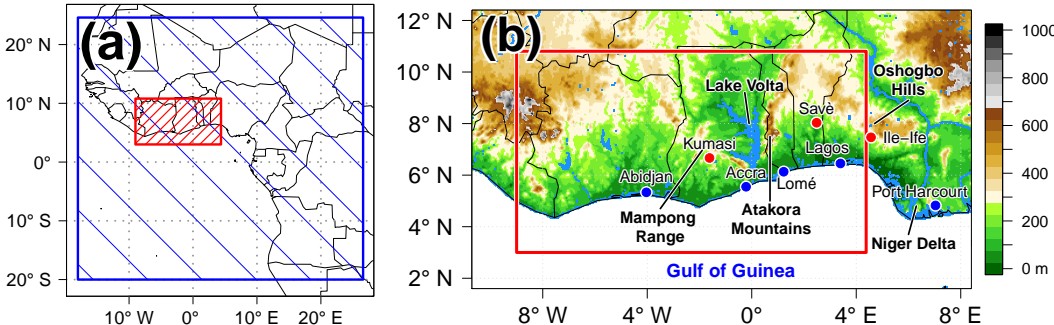

**Figure 1.** (a) Modeling domain SWA (red rectangle, 2.5 km grid mesh size) together with its coarse domain (blue, 5 km grid mesh size). (b) Map of the research area SWA. The color shading denotes the topography (m above sea level, ASL). Topographic features are named in bold, coastal cities are shown as blue dots and the three DACCIWA supersites as red dots. The modeling domain SWA is again denoted with red rectangle.

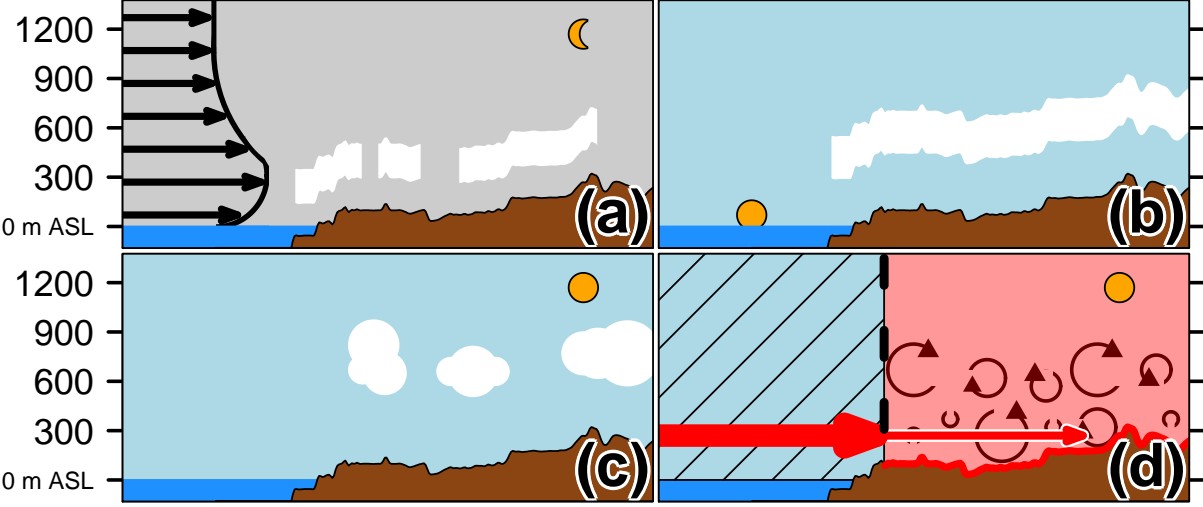

**Figure 2.** Schematic view on SWA atmospheric dynamics via a meridional-vertical transect (m ASL) through the Gulf of Guinea (blue shading) and adjacent land (brown shading). (a) During nighttime the NLLJ leads to a wind maximum at about 300 m AGL as emphasized by the black arrows. Over land, NLLS forms at the level of the NLLJ axis. (b) The maximum spatial coverage of NLLS is reached in the morning hours. After sunrise a lifting of the cloud base height can be observed. (c) During late morning or early afternoon the NLLS deck breaks up to cumuliform clouds. (d) During daytime the momentum of the onshore monsoon flow (bold red arrow) is mixed vertically over land due to atmospheric turbulence from solar heating (eddies). The balance between the monsoon flow and the turbulence leads to a frontal structure inland from the coast (black dashed line).




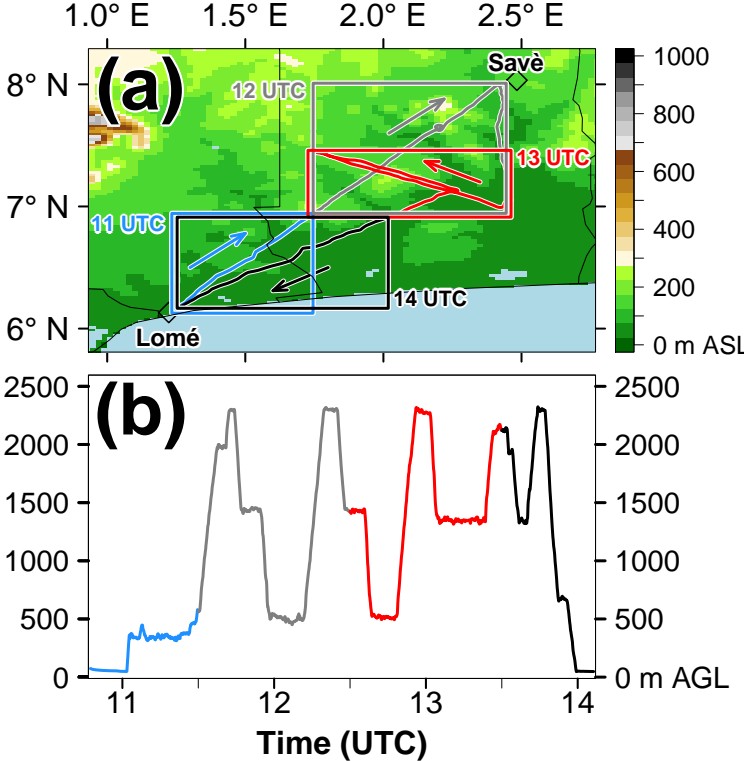

**Figure 3.** Flight track of the Twin Otter aircraft on 3 July 2016 between 10:47 UTC and 14:06 UTC (flight number TO-02) in (a) horizontal and (b) vertical dimension (m AGL). For (a) the topography (m ASL) is added. The flight track in (a) and (b) is separated in hourly time steps for the subsequent collocation with hourly model data from COSMO-ART, highlighted by the blue (10:47–11:30 UTC), gray (11:30–12:30 UTC), red (12:30–13:30 UTC) and black color (13:30–14:06 UTC). The rectangles, spanned by the horizontal extension of the hourly flight sections, are used for the selection of model data. Furthermore, the arrows in (a) indicate the flight direction with the takeoff at Lomé, the flight to Savè and the return to Lomé airport. Shortly after 12 UTC (with a flight altitude of 0.5 km AGL) the Twin Otter reached Savè. Note the meridional compression of the map in (a).





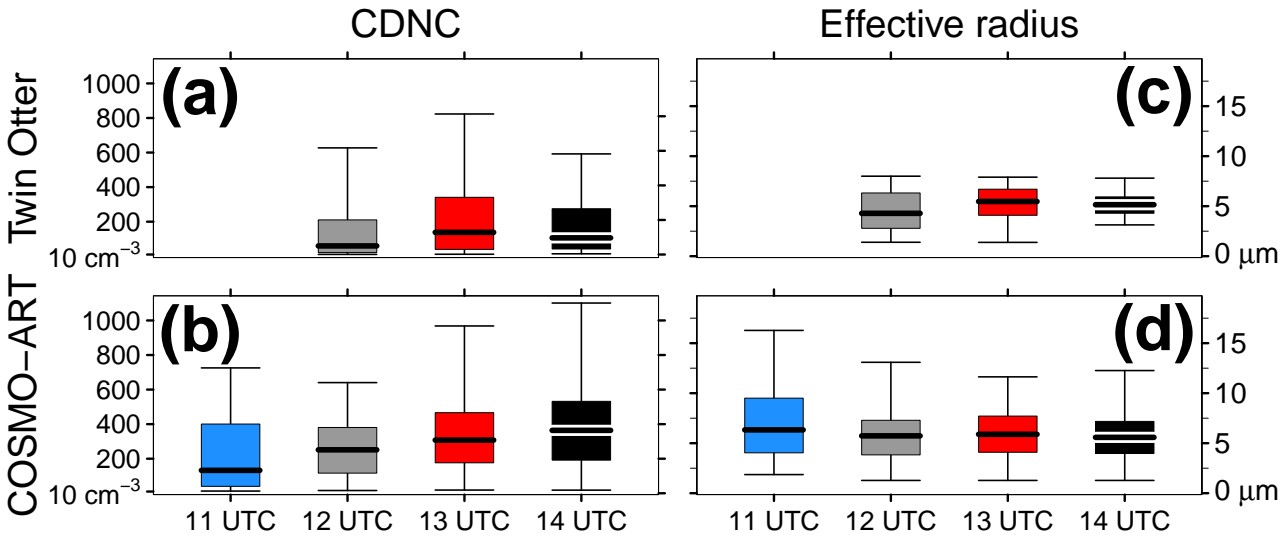

**Figure 4.** Boxplots of (left) the CDNC (cm$^{-3}$) and (right) the cloud droplet effective radius ($\mu$m) according to the flight track denoted in Figure 3a for (top) Twin Otter observations and (bottom) COSMO-ART reference case. The flight track is separated in hourly time steps from 11 UTC to 14 UTC as highlighted by the colors (compare Fig. 3). Regarding (a,c) the observations according to the flight track section with 1 s temporal resolution of the CDP device are used. Regarding (b,d) the simulation results of the cube that is spanned horizontally by the rectangles in Figure 3a and vertically about 2.3 km (in agreement with the Twin Otter maximum flight altitude) are considered. The whiskers capture the data from the 0.025 to the 0.975 quantile (95% of the data). Data outside this range are not shown. CDNC below 10 cm$^{-3}$ were omitted, leading to an observational data basis of 34, 443, 403 and 115 observations at 11, 12, 13 and 14 UTC, respectively. Due to the low observational data coverage 11 UTC is omitted.



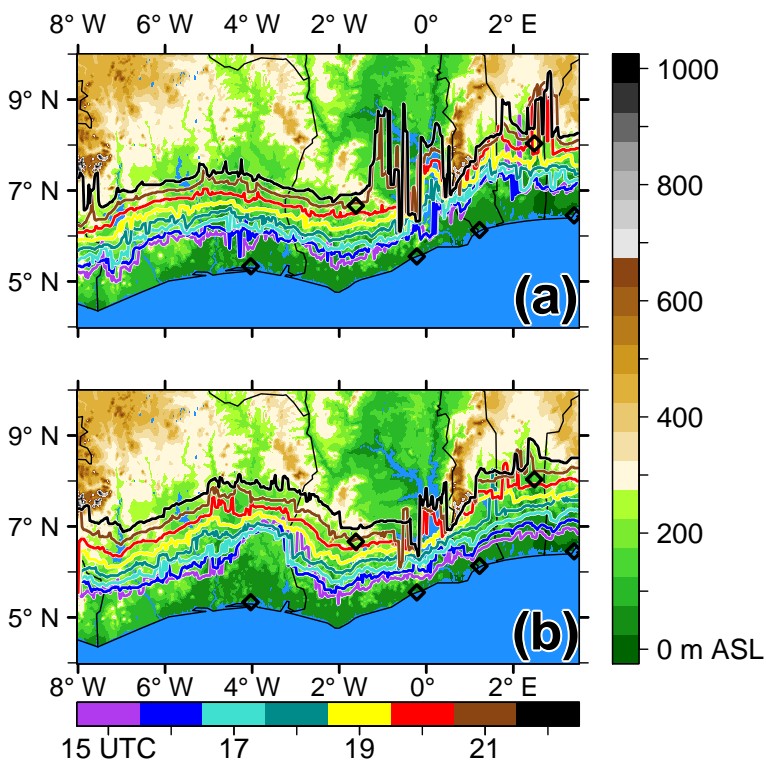

**Figure 5.** Localization of the AI front on (a) 2 July 2016 and (b) 3 July 2016 between 15 and 22 UTC for the reference case. The front is detected by the arrival of the isentropic surface $\theta_s$=302 K at $h_s$=250 m AGL. The color of the front denotes the time (UTC, bottom legend). The underlying shading shows the topography of SWA (m ASL, legend on the right). The black diamonds denote the cities shown in Figure 1.





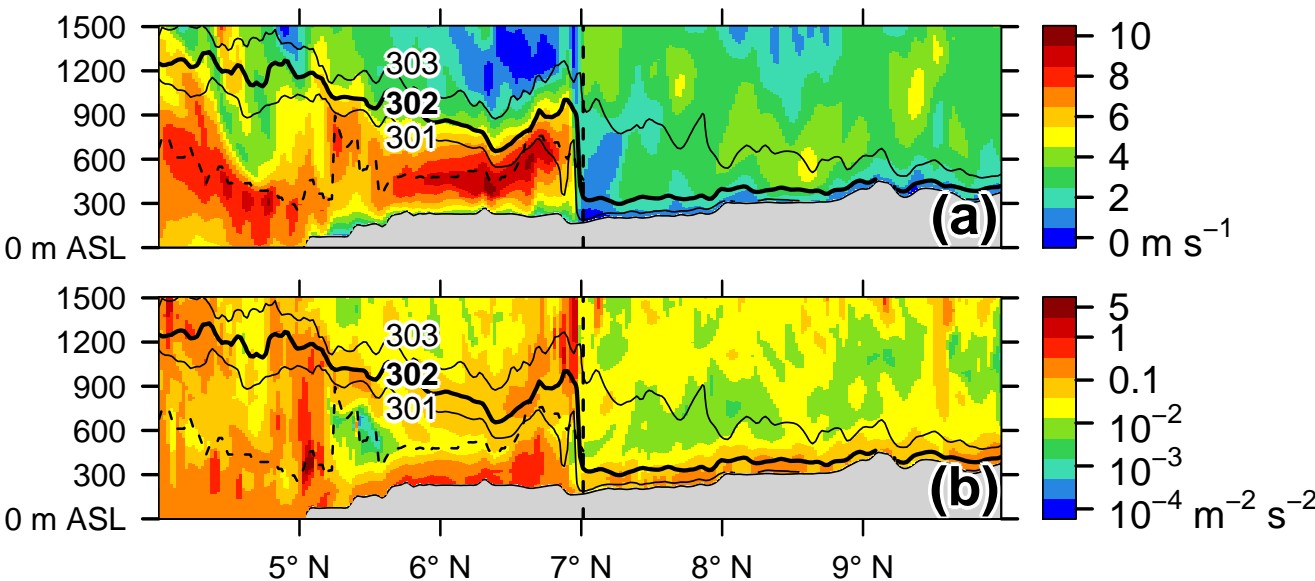

**Figure 6.** Meridional vertical transects (m ASL) of (a) wind speed (shading, m s$^{-1}$) and (b) Turbulent Kinetic Energy (TKE, m$^{-2}$ s$^{-2}$ in logarithmic scale) along 5.75°W (central Ivory Coast) for 2 July 21 UTC with respect to the reference case. The solid black contours show the potential temperature for 301, 302 and 303 K while the bold isentrope (302 K) is used for the identification of the AI front (vertical dashed line). The horizontal dashed line shows the NLLJ wind speed maximum (jet axis) in the AI post-frontal area. The gray shading indicates the topography.

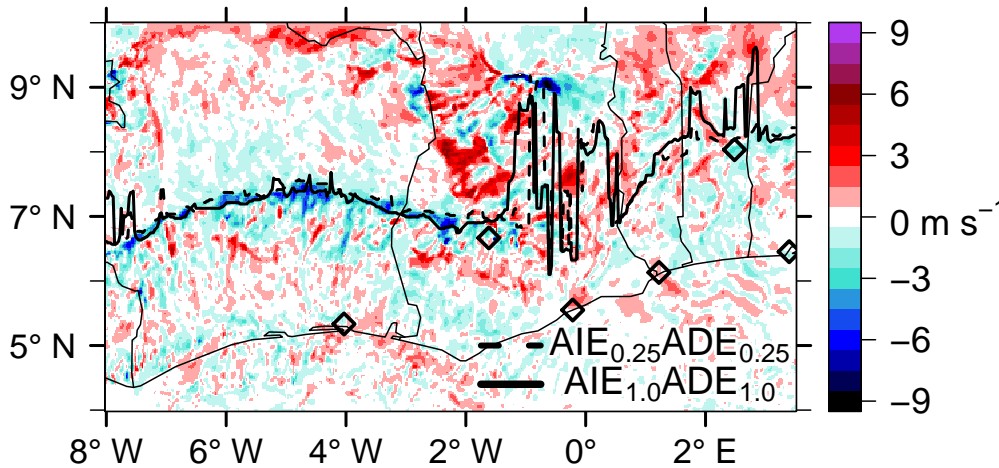

**Figure 7.** Wind speed difference at 250 m AGL on 2 July 22 UTC (m s$^{-1}$) between the reference and the clean case. The black dashed (solid) line shows the AI front for the clean (reference) case.



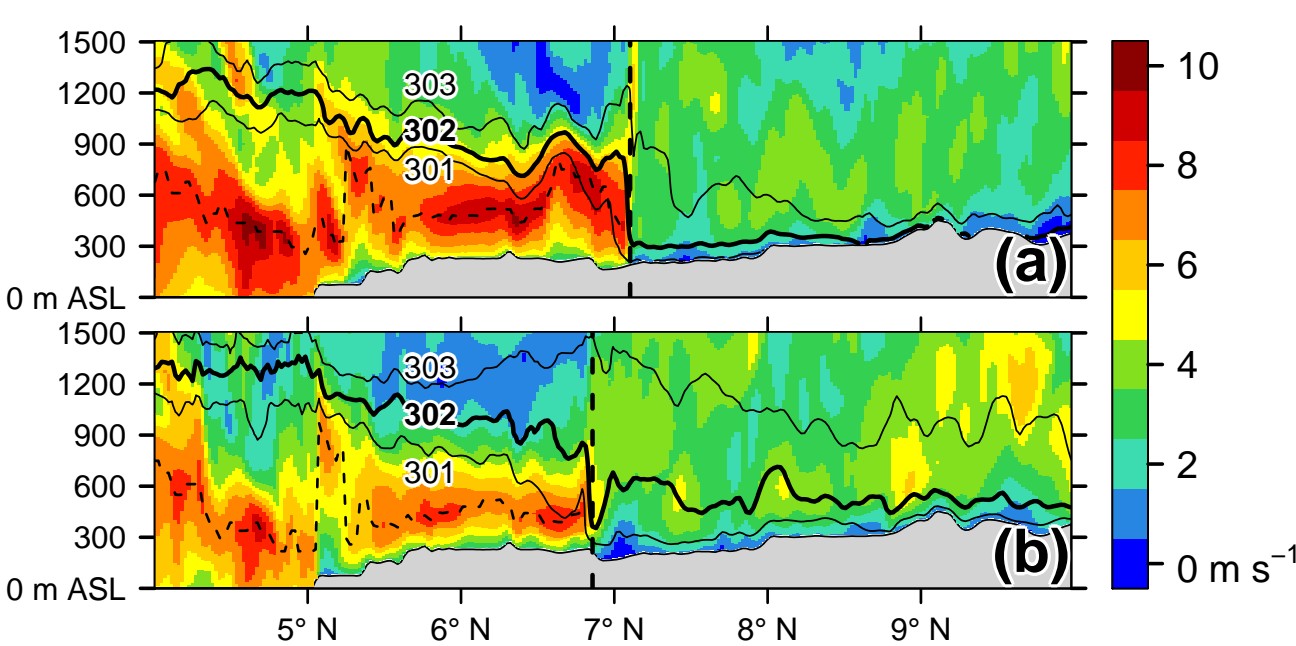

**Figure 8.** Same as Figure 6a but for (a) clean and (b) polluted case.





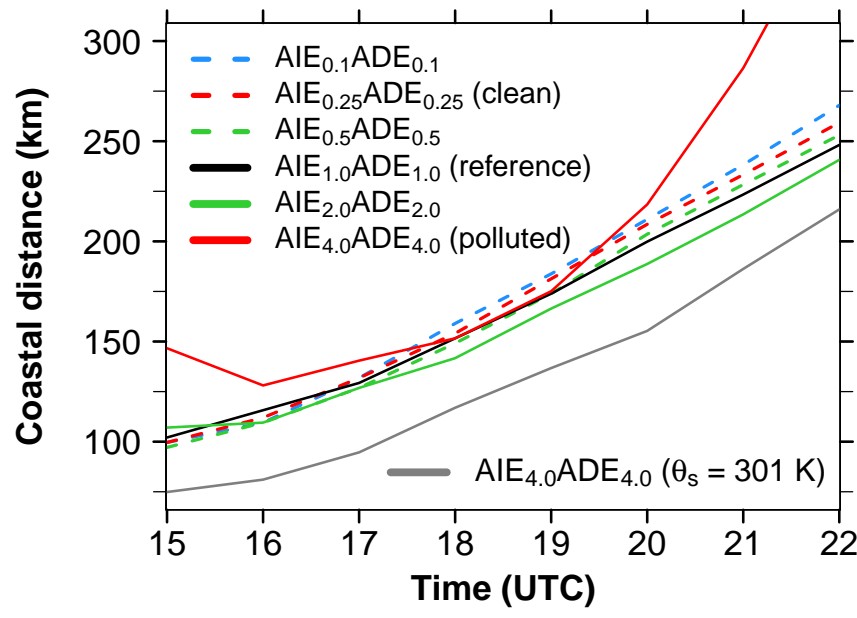

**Figure 9.** Temporal evolution of the inland propagation of the AI front via the distance from the coast (km) on 2 July 2016 between 15–22 UTC, spatially averaged over Ivory Coast (7.5–3.0°W) for the six experiments of Table 1. Dashed lines denote realizations with aerosol amounts below that of the reference case (black solid). The grey line shows the frontal propagation of the polluted case by using $\theta_s$=301 K instead of 302 K for the front detection.



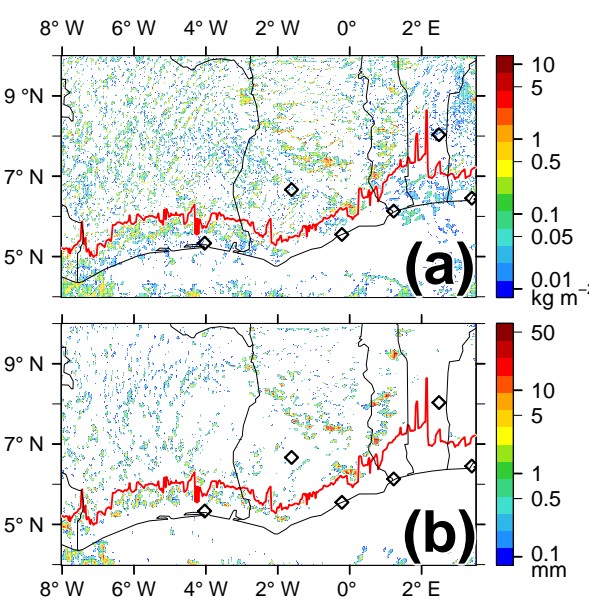

**Figure 10.** (a) Total cloud water (kg m$^{-2}$) and (b) precipitation (mm) on 2 July 15 UTC for the reference case. The red line denotes the AI front.

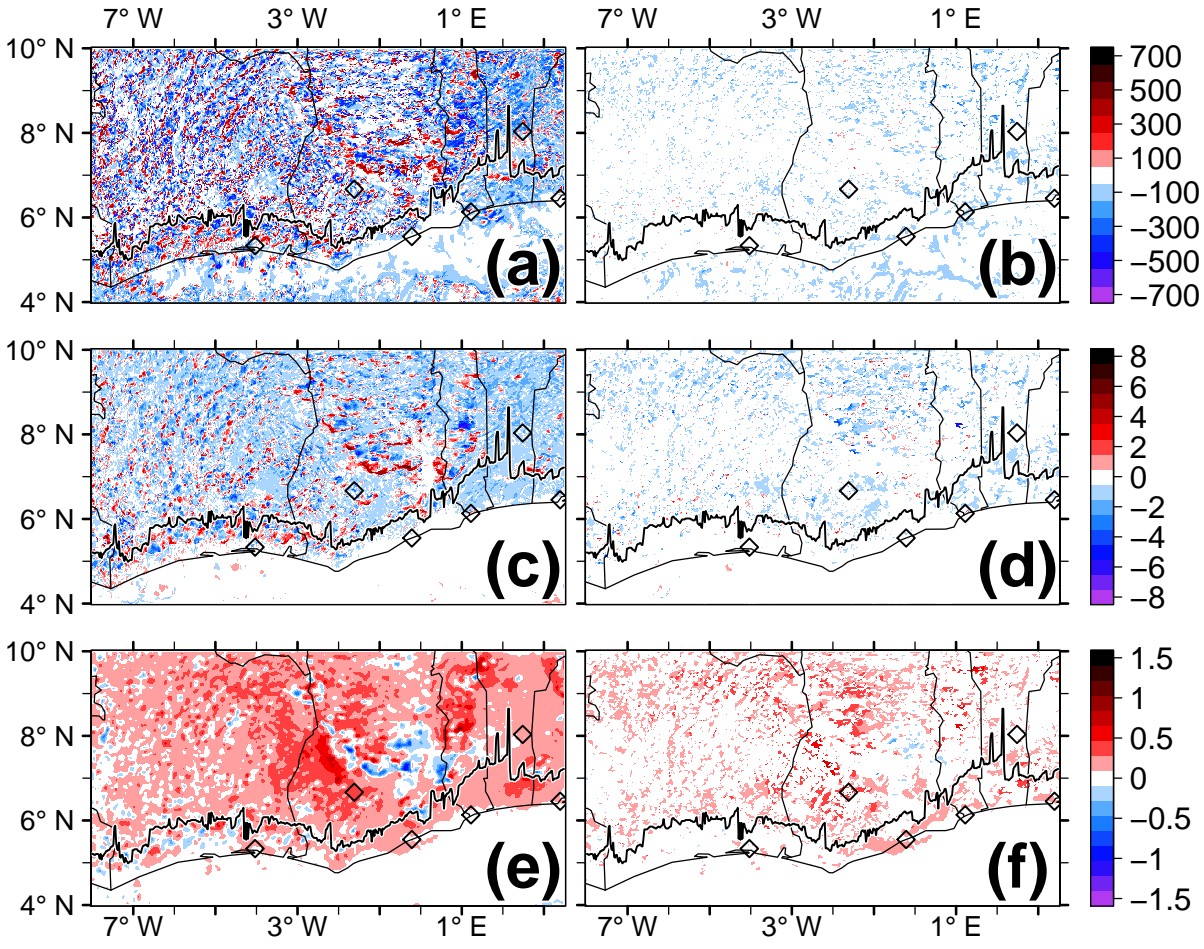

**Figure 11.** Surface meteorological quantities over SWA on 2 July 15 UTC as differences between the reference and the clean case ($AIE_{1.0}ADE_{1.0}$ - $AIE_{0.25}ADE_{0.25}$), (left) including cloudy and cloud-free areas and (right) over areas that are simultaneously cloud free in the clean and reference case. (a-b) Surface net downward shortwave radiation difference (W m$^{-2}$), (c-d) 2-m temperature difference (K) and (e-f) sea level pressure difference (hPa). The black solid lines denote the location of the reference case AI front.





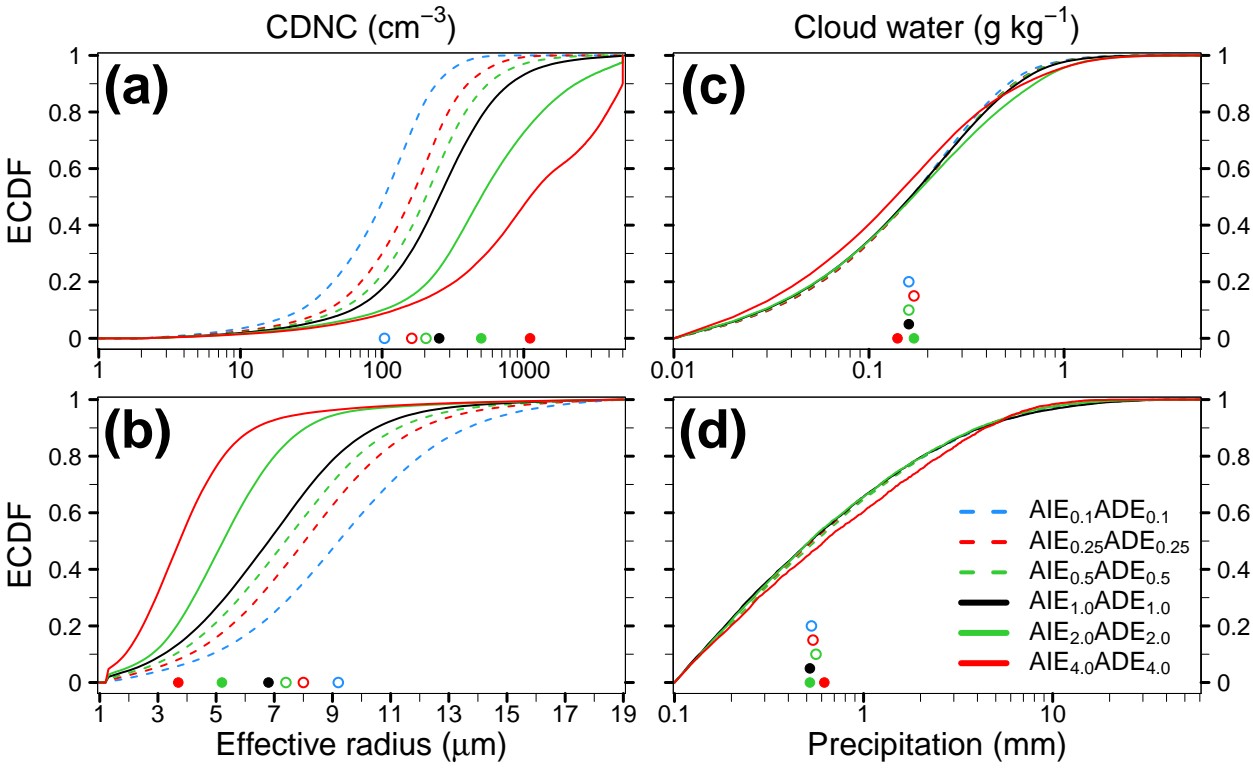

**Figure 12.** Empirical Cumulative Distribution Function (ECDF) of (a) CDNC ($cm^{-3}$), (b) cloud droplet effective radius ($\mu$m), (c) cloud water (g $kg^{-1}$) and (d) precipitation (mm) for the six experiments of Table 1 considering the full vertical column over the inland area of SWA on 2 July, 15 UTC. The circles and dots highlight the median values. Dashed lines and circles relate to realizations with less aerosol than the reference case and solid lines and dots refer to simulations with aerosol amounts greater/equal the reference case.




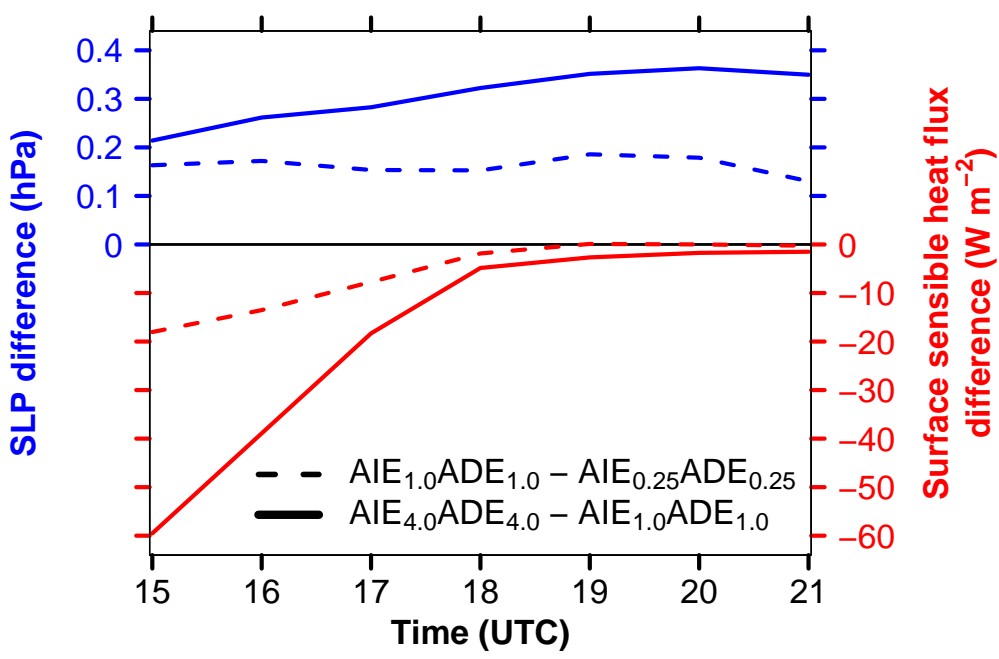

**Figure 13.** Temporal evolution of the differences in surface sensible heat flux (red, W m$^{-2}$) and surface pressure (blue, hPa) between the reference and the clean case (dashed line) and between the polluted and the reference case (solid line) for the time period 2 July 15-21 UTC spatially averaged for the AI pre-frontal area over Ivory Coast as defined by the $\theta_s$ method. The sensible heat flux is defined positive downward.





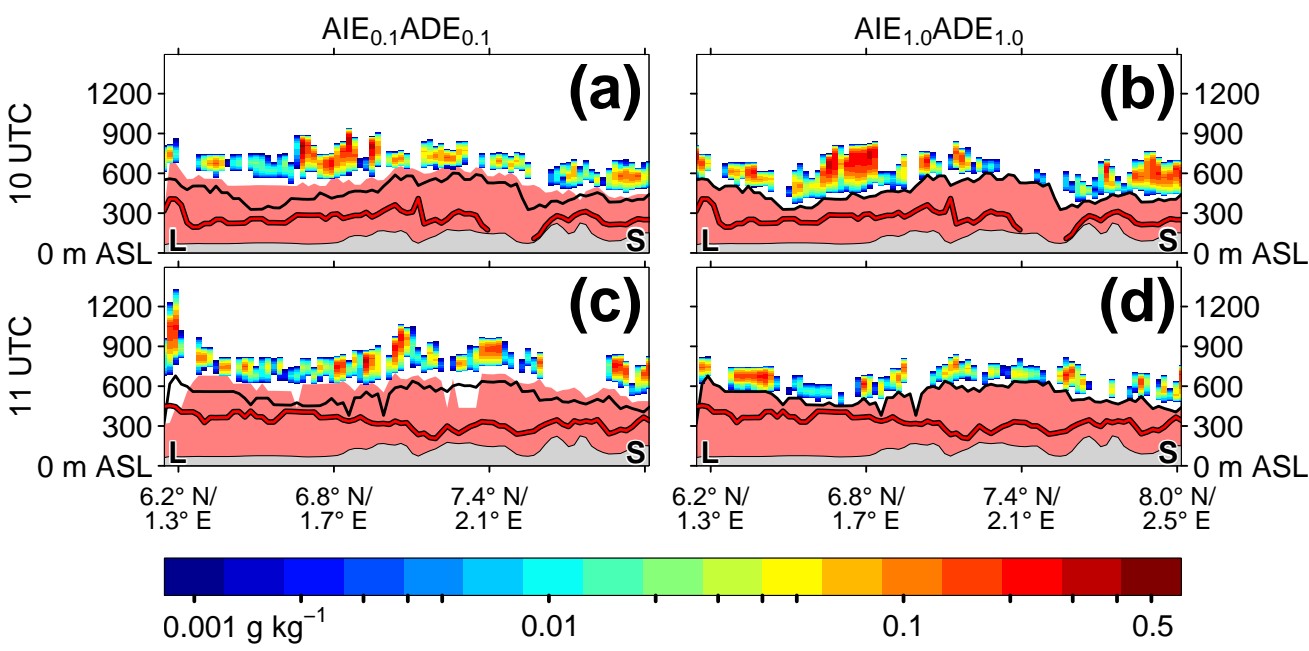

**Figure 14.** Cloud water (g kg$^{-1}$, shading) along the Lomé (L) - Savè (S) vertical transect (m ASL) for the temporal evolution on 3 July (top) 10 UTC and (bottom) 11 UTC, considering (left) the clean case and (right) the reference case. The red shading reflects instability (d$\theta$/dz<0) to highlight the evolution of the CBL. The black solid (red solid) line denotes the height of the CBL in the reference case (polluted case), simultaneously added to the panels on the left and right hand side.





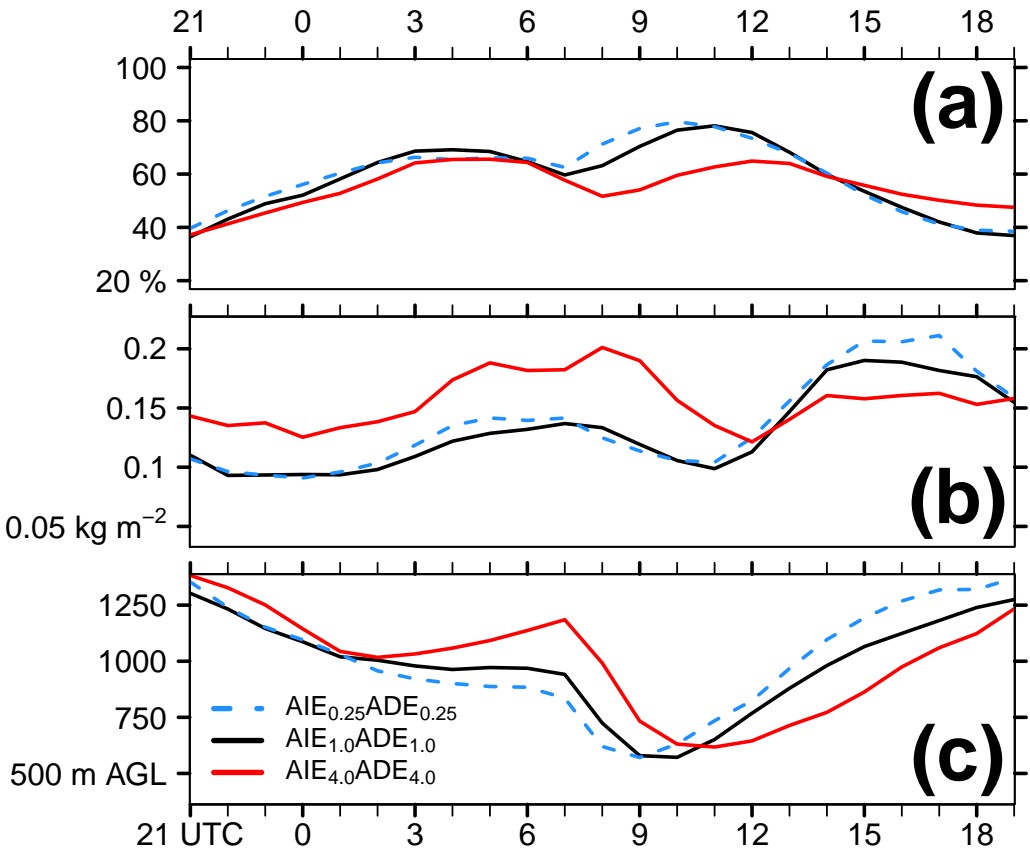

**Figure 15.** Spatial average (8°W–3.5°E, 5–10°N) of (a) total cloud cover (%), (b) total cloud water (kg m$^{-2}$) and (c) cloud base height (m AGL) for the temporal evolution between 2 July 21 UTC and 3 July 19 UTC. The cloud cover is detected by non-zero values of total cloud water. A value of 60 % denotes that 60 % of the domain is covered by clouds. For the spatial average of total cloud water, values below 10 g m$^{-2}$ were omitted. The cloud base height is detected via the lowest height AGL with a non-zero cloud water value. The blue dashed, black solid and red solid lines denote the clean, reference and polluted case, respectively.





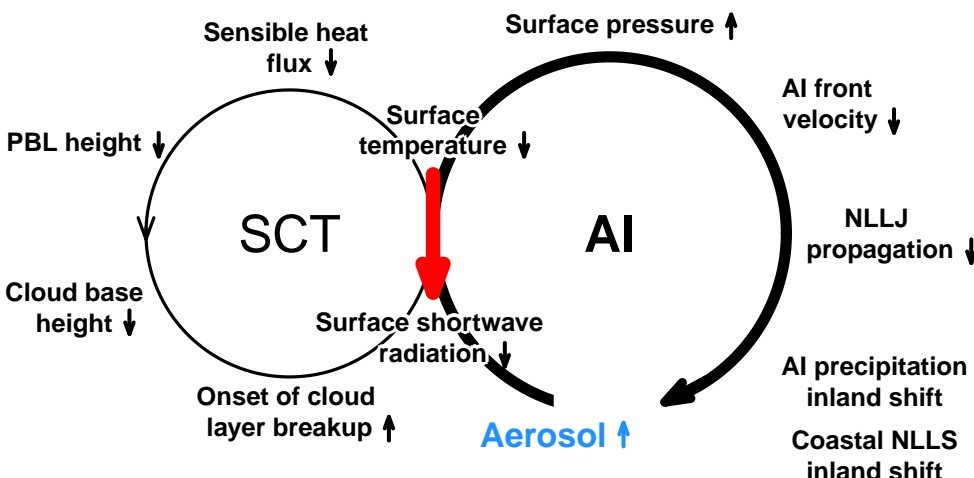

**Figure 16.** Scheme of the aerosol-related atmospheric feedbacks summarizing the findings of the process study simulations on 2–3 July 2016. The main loop is labeled AI (Atlantic Inflow) and the additional loop SCT (Stratus-to-Cumulus Transition). The small arrows in upward and downward direction denote whether a quantity reacts with a decrease (downward) or increase (upward) to the increase of aerosol mass and number (blue) as the initial perturbation. The red arrow shows the linkage between AI and SCT via the decrease in shortwave radiation and surface temperature and a potential pathway for a negative feedback of SCT on AI.





**Figure 17.** Meridional vertical transects (m ASL) of wind speed (shading, m s$^{-1}$) along 5.75°W (central Ivory Coast) for 2 July (a) 11 UTC, (b) 12 UTC, (c) 13 UTC, (d) 14 UTC and (e) 15 UTC for the reference case. The solid black contours show the potential temperature for 301, 302 and 303 K, while the bold isentrope (302 K) is used for the identification of the AI front (vertical red dashed line in (e)). The gray shading indicates the topography.



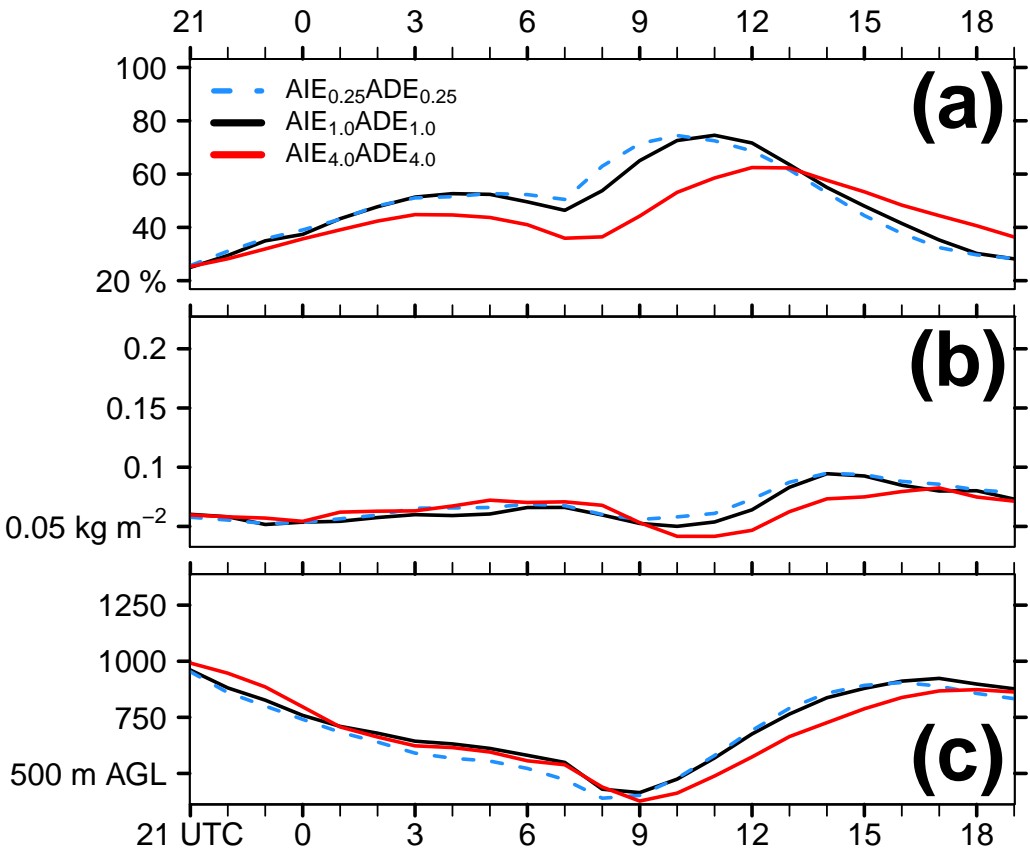

**Figure 18.** Spatial average (8°W–3.5°E, 5–10°N) of (a) cloud cover (%), (b) cloud water (kg m$^{-2}$) and (c) cloud base height (m AGL) for the temporal evolution between 2 July 21 UTC and 3 July 19 UTC with respect to clouds below 1500 m AGL. The cloud cover is detected by non-zero values of cloud water. A value of 60 % denotes that 60 % of the domain is covered by clouds. For the spatial average of cloud water, values below 10 g m$^{-2}$ are omitted. The cloud base height is detected via the lowest height AGL with a non-zero cloud water value. The blue dashed, black solid and red solid lines denote the clean, reference and polluted case, respectively.