# Peer review of "Numerical simulations of aerosol radiative effects and their impact on clouds and atmospheric dynamics over southern West Africa"

_Atmospheric Chemistry and Physics, 2018_

## Referee Comment (RC1) · Anonymous Referee #1 · 13 Apr 2018

Overview: The authors analyze six simulations of the COSMO-ART regional model with an outer domain of roughly 40 x 40 degrees centered around the Bight of Benin and an inner domain of roughly 10 x 15 degrees aligned along the Gold Coast of Southern West Africa. A main strength of the study is that the authors conduct an extensive analysis of the simulations to suss out patterns of response to aerosol conditions, and to draw some conclusions about some mechanisms and hypotheses about others. A main weakness of the study is that almost no comparisons to observations are made nor are the realism of assumed meteorological or surface or aerosol properties discussed, leaving the reader necessarily uncertain as to the degree to which the simulations are basically realistic, either in the baseline state or in the dynamic

range of aerosol conditions investigated. Especially in today's satellite age, it should not be considered a sufficient evaluation of a regional model simulation to compare results only to droplet number concentration observations. Based on the simulations, the authors advance a schematic diagram of how near-coastal meteorological conditions could be impacted by increasing regional pollution during the monsoonal period when no land-sea breeze period occurs. Observational work, for instance using satellite observations, would be required to confirm the robustness of the proposed mechanism and its strength for a given dynamic range of aerosol relative to other regional-scale drivers that are held fixed in the current study, such as sea surface temperature.

Major comments

1. In the introduction the authors refer twice to "convective-cloud invigoration mechanism," the first time apparently referring to cold clouds and the second time to warm clouds (page 2, line 32). Is this the same mechanism? Please clarify in the text to what degree the mechanism being referred to operates in simulations and under what conditions, versus established in observations and under what conditions.

2. The six simulations vary only aerosol mass and number concentrations, but how this is done is not described. The authors state that the mass and number are scaled by factors of 0.1, 0.25, 0.5, 1, 2 and 4. Since there is "aerosol-chemistry spin up" the only way I can understand this is if the values are scaled only when some process rates are calculated, but which process rates? Please clarify in the text.

3. Please report aerosol properties that correspond to the simulations somehow in Table 1 or similar format. Did the aircraft campaign for this special issue make any aerosol measurements at all that are relevant for comparison? Can the simulated aerosol conditions be compared somehow and somewhere to measurements? I consider it mandatory to indicate in the manuscript in quantitative terms (beyond a multiplicative factor) what is the dynamic range considered in this study in terms of basic measurable units such as CN, CCN, AOD, PM1, PM2.5 or the like.

4. Owing to the leading role of direct effect, simulated single-scattering albedo should be somehow quantitatively reported from simulated values and compared to measurements or other at a minimum reported simulations somewhere relevant in Africa.

5. The authors seem to focus on sensible heat flux without considering the role of soil moisture and latent heat flux (e.g., in the abstract and conceptual diagram). Is latent heat flux irrelevant at this location? Also at locations of previous studies? I have to assume that precipitation within the inner domain is negligible during the monsoon season and the surface starts out very dry, but that is not stated (please clarify in the text).

6. Please clarify in the text how soil moisture is initialized in the simulations, whether results are sensitive to how that is done.

7. Please report whether simulations are sensitive to other factors, including inner or outer domain locations or sizes, grid mesh resolution, and boundary layer turbulence or cloud schemes.

Minor corrections

1. page 2, line 19: "react" –> "are" or other fix

2. page 3, line 1: "dependent" –> "dependence" or other fix

3. recommend to divide section 5 text up from one long paragraph currently

4. recommend to guide the reader more graphically in following the transition from figure 2 (schematic diurnal cycle) to later figures (all in UTC), such as by indicating UTC time range on the panels of figure 2

---

## Referee Comment (RC2) · Anonymous Referee #2 · 16 Apr 2018

The authors focus on South West Africa, a region which is in a developing phase with an expected massive population growth and urbanisation. Therefore, an increase in anthropogenic aerosol concentration is expected. The authors assess the implication of aerosols and their possible changes on clouds and atmospheric dynamics. They present a process study with the regional model COSMO-ART. In particular, they discuss the impacts of aerosols on the propagation of the Atlantic Inflow frontal location and the Stratus to Cumulus Transition. In general, the paper is well written and the topic is of general interest. Deetz et al. conducted a detailed analysis of the performed simulations to understand the processes how aerosols influence prominent South West African dynamical features. Their main conclusions are based on three different simu-

lations, the reference, the polluted and the clean case.

I miss the discussion about the realistic representation of the current aerosol distribution in the model. Since an extensive measurement campaign took place during July 2016 it should be possible to evaluate the simulated distribution of aerosols against more measurements (here only a comparison with measures liquid cloud properties are shown).

Since the direct aerosol effect depends mainly on the radiative properties of the aerosols it is of interest to show the aerosol composition in the region during the 2nd -3rd of July. And again, it would be helpful if the simulated aerosol radiative properties or the simulated radiative fluxes could be evaluated against observations.

To understand the full meaning of polluted and clean case it is necessary to know about the aerosol content and composition in the reference case. Without that knowledge, a fractional increase or decrease is not meaningful. Do the authors change the aerosol concentration of the different types equally? This should be clarified in the revised manuscript.

Another clarification is needed in terms of the general model setup. How are aerosols treated at the outer boundaries? Are they prescribed by output of global model simulations? The meteorological state is initialized every day at 0 UTC. Are the wind and temperature fields pulled back to the ICON forecast every day at 0 UTC? If yes, how is it possible to analyse the impact of the direct and indirect aerosol effect on the dynamics? I also wonder about the choice of the inner model domain (figure 1, indicated by red box). The western as well as the eastern and part of the northern boundary are located in a mountainous region. Could that cause problems due to resolution effects?

Minor comments:

Page 1 line 22: The population is expected to growth.

Page 2 line 19: Please replace "react" with "are".

[Figure]

Page 10 line 25: Please rewrite that sentence

Table 1: I recommend to rename the simulations (the names are unnecessary long), ADE and AIE are scaled by the same factor, the simulations could be named as AE0.1, AE0.25... (AE = aerosol effect)

General remark: Maybe it is not necessary to present results of all 6 simulations. It underlines somehow the results but for the discussion it seems not to be important to present them. I recommend that the authors rethink the demand to present the AE0.1, AE0.5, AE2 simulations in the paper.

---

## Referee Comment (RC3) · Anonymous Referee #3 · 17 Apr 2018

In this study, the authors ran a regional climate model COSMO-ART at convection-perming resolutions to examine the effects of aerosols on weather and climate based on a 2-day case study. They documented in detail how aerosol affects cloud and atmospheric dynamics over southern West Africa. They further presented detailed analysis of mechanisms that leads to these changes, and provide a conceptual model for this. I think overall the paper is well written, and it is great addition to existing literatures on aerosols effects on climate over Africa. I would recommend it publication after my following comments are addressed:

Major comments: 1. I understand these are expensive simulations, but I still think it

would be really nice if the authors can run model longer, say a month. The current results are interesting, but it is less clear how robust these results are. A longer simulation would definitely be more interesting, and may also produce more robust results. 2. Their model does have the capability to separately treat AIE and ADE. But in the paper, the authors examined the two effects together. Separating these two may help to answer whether AIE or ADE dominates in this case study.

Specific comments: Title: The paper is about aerosol effects on atmospheric dynamics in a case study. But the title said "cloud and aerosol radiative effects as key players for anthropogenic changes in atmospheric dynamics over southern West Africa". I think the title is misleading and confusing. Firstïij Ňthe paper is not about cloud radiative effectsïijŇthough it does talk about aerosol radiative effects through its impact on clouds. But this is different from cloud radiatve effects. Second, the paper only documents aerosol effects on atmospheric dynamics based a case study from model simulations. "anthropogenic changes in atmospheric dynamics" may sound like this is what you observed. As this effect is purely a modeling study, I suggest the authors to clarify this in the title.

Section 2.1: model experiments and AIE. It looks like the authors can separately examine the effects of AIE and ADE, but in all model experiments documented here, AIE and ADE are examined together. If the authors examine AIE and ADE separately, this may help to clarify some points the authors made regarding the relative roles of AIE and ADE on SWA. This relates to some of the discussions in Section 6 (e.g., the last paragraph).

Section 5: the first paragraph is overly long. Suggest to separate it into several short paragraphs with a focus theme in individual paragraphs.

Page 9, line 21: what are these two numbers? The same question is also applied for next three lines (lines 22-24)

Page 10, line 4-5: an aerosol increase has large impacts than the aerosol decrease.

This is a little bit surprise to me. I would expect when aerosol concentrations further increases, its effects saturate, and its effects decreases (e.g., numerous small particles compete for water vapor so a lower maximum supersaturation is expected). So can you elaborate what might happen here.

Page 10, lines 12-14: The Twomey effect is also through changes in cloud optical thickness, but not through cloud water. So the second half of this statement is confusing.

Page 10, lines 28-29: this statement is not clear to me ("it is interesting that . . .").

Page 10, the pressure gradient mechanisms: Here sea surface temperature was not affected by aerosol loading. So this overestimates the effects of aerosols on land-sea temperature differences. Any discussion on this?

———————————————————

---

## Author Comment (AC1) · 19 Jun 2018

Answer to Referee #1 Konrad Deetz 19 June 2018

Dear Referee (Atmospheric Chemistry and Physics),

thank you for your report from 13 April 2018. We have accounted for the comments and suggestions in the revised manuscript version. Please find our replies (marked with #) to the individual comments in the following. Before the detailed replies to your comments we want to stress one important overarching point: This study mainly focuses on the sensitivity of atmospheric dynamics and cloud properties to aerosols and not on a

detailed validation of the model system. Nevertheless, we have done a comprehensive evaluation of the model with the available observations of the DACCIWA measurement campaign. We show corresponding figures in our replies which appear at the end of this document. (The complete figure captions are given within the text because the figure caption space for the uploaded figures is not sufficient.)

Sincerely, Konrad Deetz on behalf of all coauthors

Referee comments:

(0) The authors analyze six simulations of the COSMO-ART regional model with an outer domain of roughly 40 x 40 degrees centered around the Bight of Benin and an inner domain of roughly 10 x 15 degrees aligned along the Gold Coast of Southern West Africa. A main strength of the study is that the authors conduct an extensive analysis of the simulations to suss out patterns of response to aerosol conditions, and to draw some conclusions about some mechanisms and hypotheses about others. A main weakness of the study is that almost no comparisons to observations are made nor are the realism of assumed meteorological or surface or aerosol properties discussed, leaving the reader necessarily uncertain as to the degree to which the simulations are basically realistic, either in the baseline state or in the dynamic range of aerosol conditions investigated. Especially in today's satellite age, it should not be considered a sufficient evaluation of a regional model simulation to compare results only to droplet number concentration observations. Based on the simulations, the authors advance a schematic diagram of how near-coastal meteorological conditions could be impacted by increasing regional pollution during the monsoonal period when no land-sea breeze period occurs. Observational work, for instance using satellite observations, would be required to confirm the robustness of the proposed mechanism and its strength for a given dynamic range of aerosol relative to other regional-scale drivers that are held fixed in the current study, such as sea surface temperature.

**The high resolution realizations with COSMO-ART are computational expensive and**

therefore we had to restrict to a short time period. We commonly agreed on the 3-4 July 2016 as a golden day due to the intensive stratus period observed over Save (Kalthoff et al., 2018). The data analysis of the COSMO-ART results revealed that the AI is most pronounced and coherent over Ivory Coast (Figure 5 in the manuscript). Therefore, we decided to focus on this area although we are aware of the fact, that the DACCIWA measurement campaign with its supersites and aircraft operations focus more on the eastern part of the domain. For 3-4 July, no aircraft observations are available for the coastal region of Ivory Coast. Furthermore, a detailed assessment of the MODIS AOD data revealed that the data availability over SWA (land area) is substantially reduced by the presence of clouds (see Review-Figure 6). Nevertheless, we comprehensively evaluated COSMO-ART in (a) past studies and also (b) within DACCIWA for SWA. (a) The full capacity of COSMO-ART was applied in numerous studies. Knote et al. (2011) validated the aerosol and gaseous compounds in detail against observations for the European area.Stanelle et al. (2010) analyzed the ADE of mineral dust over the Sahara that alters the near-surface temperature up to 4 K in case of elevated mineral dust layers. Furthermore, feedbacks between the mineral dust ADE and the atmospheric dynamics lead to modifications in the mineral dust emission. Athanasopoulou et al. (2014) quantified a severe wildfire event over Greece in 2007 in terms of air quality and ADE that reveals AOD values between 0.75 and 1 and a cooling of 0.5 K.Walter et al. (2016) extended COSMO-ART with a plume-rise model to describe biomass burning pollution injection heights and applied the model to Canadian forest fires in 2010. The ADE related to the biomass burning plume leads to a near-surface cooling of up to 6 K. The ADE of sea salt over the Mediterranean Sea, Northeast Atlantic, North Sea and Baltic Sea was modeled by Lundgren et al. (2013) in accordance to the observations from remote sensing. Kraut (2015) applied an ensemble approach by including random temperature pertubations to isolate the sea salt AIE on the characteristics of a cyclone over the Mediterranean Sea in 2011, revealing spatial shifts and intensity differences in the precipitation patterns. By considering the AIE on post-frontal convective clouds over Germany in 2008, the cloud properties were changed significantly, leading to a

reduction in precipitation with increasing aerosol amounts (Rieger et al., 2014). (b) In the preparation of the high resolution process study simulations with COSMO-ART for SWA, we conducted operational forecasts for the area over the time period 8 March to 31 July with a grid mesh size of 28 km. This allows us to comprehensively analyze the model performance with respect to meteorology and air pollution and to prepare reasonable COSMO-ART data for the nesting of our high resolution realizations. Also we found tendencies of overestimations of trace gas concentrations in COSMO-ART, likely due to uncertainties in the emission inventories, COSMO-ART reasonably reproduces the SWA meteorological and air pollution characteristics. To support these findings and to meet the concerns of the reviewer, the following 7 figures are attached to this review answer:

- Review-Figure-1: Temporal evolution of the height (m AGL) of the wind speed maximum between 0 and 1500 m AGL for the mean 57 h forecast lead time. (13 June - 30 July 2016) at Savè as observed (black, Doppler Lidar) and modeled with COSMO-ART (blue). The shaded areas denote the standard deviation.

- Review-Figure-2: Wind speed profile (m s-1) between 0 and 2000 m ASL as mean diurnal cycle (13 June - 30 July 2016) at Savè for (a) COSMO-ART and (b) Doppler Lidar observation.

- Review-Figure-3: Vertical profiles (km AGL) of BC (mg m-3) at Savè for (a-c) 5 July 2016, (d-g) 14 July 2016 and (h-i) 15 July 2016. The ALADINA (small unmanned aircraft) observations of total BC are denoted in black, the COSMO-ART results for fresh BC, aged BC (Aitken mode), aged BC (accumulation mode) and total BC are shown in green, blue, brown and red, respectively. The observations were temporally assigned to the 3 hourly model output with a deviation not greater than 1 hour and by subsequently interpolating the model data to the ALADINA altitudes. Within these time steps, ALADINA conduct several ascends and descends. It is assumed that the observations within the time steps are measured instantaneously and the data is sorted according to their altitude, to allow for clearness of the visualization.

- Review-Figure-4: Comparison of Twin Otter measurement flight TO-16 (14 July 2016, 06:44 UTC to 09:50 UTC) results with COSMO-ART. For a comparison the model output of 9 UTC and the measurements 15 minutes around this time step (08:45-09:15 UTC) were selected. (a) Flight altitude (m AGL), (b) flight track, (c) NOx concentration (ppbv) at 750m height and flight track, (d) vertical transect of NOx concentration (ppbv) along the flight track with aircraft observations included, (e) temperature ($^{\circ}$C), (f) specific humidity (kg kg-1), (g) CO concentration (ppbv), (h) NO concentration (ppbv), (i) NO2 concentration (ppbv), (j) NOx concentration, (k) O3 concentration (ppbv) and (l) SO2 concentration (ppbv). The panels (e)-(l) present the COSMOART results in blue and the observations in black. The horizontal color lines on top of these panels are denoted to the colors in panel (a) and (b) to illustrate the aircraft location related to the observed trace gas concentrations.

- Review-Figure-5: Mean total AOD averaged from 27 June to 17 July 2016 of (a) COSMO-ART, spatiotemporally collocated with MODIS Terra, (b) MODIS Terra, (c) COSMO-ART, collocated spatiotemporally with MODIS Aqua and (d) MODIS Aqua.

- Review-Figure-6: Number of observations within the time period 27 June - 17 July of (a) MODIS Terra and (b) MODIS Aqua

- Review-Figure-7: AOD (550 nm) at Savè from COSMO-ART (blue), CAMS (green) and AERONET (red) between 13 June - 31 July 2016 for (a) mineral dust, (b) sea salt, (c) anthropogenic aerosol and (d) total aerosol. Consider the different scaling of the ordinates. Data gaps are related to technical issues during the forecast.

Further intercomparison between COSMO-ART and observations obtained within DACCIWA including the supersites and aircraft, other models or remote sensing data is summarized in Section 5 of (https://publikationen.bibliothek.kit.edu/1000077925). However, we have not added the evaluation material since it would distract from the main purpose of the study to disentangle potential effects from aerosol on AI and SCT in a sensitivity study. Nevertheless, we agree that the paper also has to show that it

is generally able to reasonably reproduce the conditions in SWA. To account for your comment, we added an intercomparison of the shortwave, longwave, sensible and latent heat flux with observations at Save supersite in Appendix A and adapted Section 2.2 (Observational data) as follows:

"Within the DACCIWA project, an extensive field campaign took place in June–July 2016 in SWA (Fig. 1b) (Flamant et al., 2018). The time period was selected to capture the onset of the WAM and a period characterized by increased cloudiness. The DACCIWA ground-based measurement campaign encompassed the time period from 13 June to 31 July 2016, including the three supersites Kumasi (Ghana), Savè (Benin) and Ile-Ife (Nigeria) (red dots in Fig. 1b). A complete overview of the DACCIWA ground-based measurement campaign, their supersites, instrumentation and a first insight into the available data is presented in Kalthoff et al. (2018). The DACCIWA airborne measurement campaign captured the time period from 27 June to 17 July 2016 (Flamant et al., 2018). The focus of this study is on Ivory Coast and therefore less observational data from the DACCIWA campaign is available for evaluation. However, a substantial evaluation with respect to meteorology and air pollution is realized with COSMO-ART over SWA with respect to other time periods and by focusing on the eastern part of the research area (not shown). This is presented in Deetz (2018). For this study, observations of the liquid cloud properties from the CDP-100 (Cloud droplet probe, data revision 3) of the British Antarctic Survey (BAS) Twin Otter aircraft on 3 July 2016 are used for a comparison with COSMO-ART. The CDP-100 is a wing mounted canister instrument including a forward-scatter optical system to measure the cloud droplet spectrum between 2-50 $\mu$m with a frequency of 1 Hz. Additionally, the comparison of the modeled net downward shortwave and longwave radiation as well as the sensible and latent heat flux with Savè supersite is presented in Figure 17 of Appendix B. COSMO-ART reasonably reproduces the fluxes with lower fluxes with increasing aerosol as expected."

Furthermore, we added an intercomparison between COSMO-ART and the ATR42

SAFIRE aircraft with respect to the aerosol number density. However, this evaluation focuses on the Lome-Save area and not on Ivory Coast (since for this date no observations for Ivory Coast are available). In Section 2.2 (Observational data) we added the following text: "The aerosol aerosol number density is evaluated using observations of the ATR42 SAFIRE (Service des Avions Français 25 Instrumentés pour la Recherche en Environnement) for the 3 July 2016. Additionally, the comparison of the modeled net downward shortwave and longwave radiation as well as the sensible and latent heat flux with Savè supersite is presented in Figure 19 of Appendix B. COSMO-ART reasonably reproduces the fluxes with lower fluxes with increasing aerosol as expected." In Section 4 (Evaluation of modeled cloud and aerosol properties with aircraft observations) we added the following text: "The research aircraft ATR42 SAFIRE also obtained aerosol properties in the Lomé-Savè area on 3 July 2016 (8:32–13:16 UTC). The flight track and altitude is presented in Figure 5, showing similar flight patterns compared to the Twin Otter (Fig. 3). By assuming dry aerosol, Figure 6 shows the comparison between COSMO-ART and the Spectrometer Scanning Mobility Particle Sizer (SMPS) to evaluate the Aerosol Number Density in the size range 0.02–0.5 mu m. Figure 6 reveals that the modeled aerosol number density shows a similar temporal evolution compared to the observations but has a constant bias, overestimating the observed aerosol number density by a factor of about 2 (indicated by the blue dashed line). Therefore, in the subsequent study it has to be considered that the reference case shows already higher aerosol concentrations compared to the current state in SWA as quantified by the aircraft measurements. Overall, the evaluation reveals that COSMO-ART is capable to reproduce the aerosol situation on 3 July 2016 over SWA which is the basis for further sensitivity studies.

The figures related to this passage are attached in this review answer:

Fig.5 -> Review-Figure-8: Flight track of the ATR42 SAFIRE on 3 July 2016 between 08:32 UTC and 13:13 UTC in (a) horizontal and (b) vertical dimension (m AGL). For (a) the topography (m ASL) is added. The flight track in (a) and (b) is separated in hourly

[Figure]

time steps for the subsequent collocation with hourly model data from COSMO-ART, highlighted by the pink (08:32—09:30 UTC), blue (09:30–10:30 UTC), gray (10:30–11:30 UTC), red (11:30–12:30 UTC) and black color (12:30–13:13 UTC). Furthermore, the arrows in (a) indicate the flight direction with the takeoff at Lomé, the flight to Savè and the return to Lomé airport. Shortly. Note the meridional compression of the map in (a).

Fig.6 -> Review-Figure-9: Aerosol number density (AND, cm-3) in the size interval 0.02 to 0.5 mu m as measured by the Spectrometer Scanning Mobility Particle Sizer (SMPS) on board the ATR42 (black) and modeled with COSMO-ART (solid blue, reference case). The horizontal dashed blue line shows the COSMO-ART AND divided by 2. The vertical blue dashed lines indicate the COSMO-ART model output hours, which are compared to the observations.

Fig. 19 (Appendix B) -> Review-Figure-10: Comparison between Savè supersite observations (grey) and COSMO-ART reference (black), clean (blue) and polluted (red) of (a) net downward shortwave radiation (W m-2), (b) net downward longwave radiation (W m-2), sensible heat flux (W m-2) and latent heat flux (W m-2). The horizontal lines in (a) denote clouds over Savè in the observations and COSMO-ART.

You mentioned that the study does not explicitly exclude the land sea-breeze but land sea-breeze effects can hardly be disentangled from AI effects because the monsoon flow superimposes the land-sea breeze. The conclusion of the manuscript closes with the suggestion of ideas to assess the aerosol-AI impact via observation. We proposed: "A potential strategy is the analysis of the AI front around noon via remote sensing cloud observations from past to present by assuming a positive trend in the aerosol burden. It is expected that the daytime AI front location has shifted landwards from the past to current conditions but also other phenomena (e.g. decadal SST variations) have the potential to affect the front location." This assessment is suitable for a companion paper but is clearly beyond the scope of this paper.

(1) In the introduction the authors refer twice to "convective-cloud invigoration mechanism," the first time apparently referring to cold clouds and the second time to warm clouds (page 2, line 32). Is this the same mechanism? Please clarify in the text to what degree the mechanism being referred to operates in simulations and under what conditions, versus established in observations and under what conditions.

**Saleeby et al. (2014) show that the convective-cloud invigoration mechanism is not restricted to cold clouds. Also in warm cumuliform clouds, the enhanced condensation under polluted conditions can lead to further release of latent heat, intensified upward motion and therefore to more clouds. This is considered by the model. I don't see your point. Could you please explain more in detail what do you expect?**

(2) The six simulations vary only aerosol mass and number concentrations, but how this is done is not described. The authors state that the mass and number are scaled by factors of 0.1, 0.25, 0.5, 1, 2 and 4. Since there is "aerosol-chemistry spin up" the only way I can understand this is if the values are scaled only when some process rates are calculated, but which process rates? Please clarify in the text.

**Generally, the aerosol amount during the whole simulation period is not changed. Just when it comes to the calculation of the radiative transfer (in case of ADE) and the aerosol activation (AIE) the aerosol mass and number is scaled. We clarified it in the text (Section 2.1): "Note, that the aerosol scaling only comes into consideration when deriving the aerosol optical properties (with respect to ADE) and the aerosol activation (with respect AIE). All aerosol dynamic processes remain unaffected by the scaling."**

(3) Please report aerosol properties that correspond to the simulations somehow in Table 1 or similar format. Did the aircraft campaign for this special issue make any aerosol measurements at all that are relevant for comparison? Can the simulated aerosol conditions be compared somehow and somewhere to measurements? I consider it mandatory to indicate in the manuscript in quantitative terms (beyond a multiplicative factor) what is the dynamic range considered in this study in terms of basic

measurable units such as CN, CCN, AOD, PM1, PM2.5 or the like.

\# Refers to (0). Furthermore we added a plot to quantify the aerosol change that is related to the aerosol scaling: - Review-Figure-11: Temporal evolution of median (a) total aerosol number (cm-3) and (b) total aerosol mass (mu g m-3) in the lowest 2 km AGL over Ivory Coast (7.5–3.0°W) between 2 July 15 UTC to 3 July 21 UTC for the clean (blue dashed), reference (black solid) and polluted case (red solid), based on the aerosol scaling introduced in Table 1.

(4) Owing to the leading role of direct effect, simulated single-scattering albedo should be somehow quantitatively reported from simulated values and compared to measurements or other at a minimum reported simulations somewhere relevant in Africa.

\# The SSA is calculated in COSMO-ART to derive the aerosol effect on radiation. However, the SSA is not a standard output variable and therefore is not available for a comparison with observations. As far as we know, SSA observations are not available within DACCIWA except of AERONET. AERONET has three relevant stations: KITcube Save, Ilorin and Koforidua. They provide the SSA but only in four discrete wavelength. COSMO-ART uses wavebands (intervals) so a direct comparison is not possible. To focus only on the time period 2-3 July 2016, as done in this study, allows no robust evaluation. The comparison of modeled and observed AOD is compared with AERONET and the CAMS model in Review-Figure-7 for a longer period. It supports our finding that the anthropogenic aerosol is overestimated in COSMO-ART likely due to the uncertainty in the emission inventories.

(5) The authors seem to focus on sensible heat flux without considering the role of soil moisture and latent heat flux (e.g., in the abstract and conceptual diagram). Is latent heat flux irrelevant at this location? Also at locations of previous studies? I have to assume that precipitation within the inner domain is negligible during the monsoon season and the surface starts out very dry, but that is not stated (please clarify in the text).

[Figure]

\# We agree on that and added the consideration of the latent heat flux in the text. According to that the latent heat flux curve is added in Figure 13 of the manuscript. See Review-Figure-12.

(6) Please clarify in the text how soil moisture is initialized in the simulations, whether results are sensitive to how that is done.

\# Since we focus on short time periods in the order of days, the COSMO-ART realizations are performed in NWP mode, the soil moisture is initialized via the meteorological boundary conditions of ICON. Therefore there is no long-term spinup of soil moisture as it is done for climate projections e.g. when using the CLM version of COSMO. NLLS is not related to significant amounts of precipitation (sporadical drizzle) and is therefore not altering the soil moisture. We expect that the cool and moist air, advected with AI, dominates the meteorological characteristics as presented in Figure 6. We agree that soil moisture is worth to focus on, when we talk about the onset of convection in the afternoon. However, the sensitivity of soil moisture on the aerosol-AI interactions is beyond the scope of our study. The soil moisture is also important when parameterizing the emission of mineral dust particles. In Deetz et al. (2016) I have shown that the soil moisture has significant impact on the mineral dust emission in Northeastern Germany. But for this SWA case study, the mineral dust contribution (primarily coming from the Sahara) is small and in this arid region the sensitivity towards soil moisture is of less importance than for central Europe.

(7) Please report whether simulations are sensitive to other factors, including inner or outer domain locations or sizes, grid mesh resolution, and boundary layer turbulence or cloud schemes.

\# As described in the answer to comment (0), we have conducted operational forecasts with COSMO-ART from 8 March to 31 July 2016 with a grid mesh size of 28 km and a large domain capturing wide areas of Africa (25W-40E,20S-35N). It reveals that this coarse setup including parameterized convection shows deficiencies in the

representation of the SWA meteorological conditions. The precipitation forecasts show less discriminance and incoming shortwave radiation and 2 m temperature are underestimated compared to Save observations. Therefore we decided not to use this data as boundary data for the nesting. We performed tests with a higher grid mesh size of 5 km, used explicit and parameterized convection, different turbulence closures available in COSMO and also tested it with and without the two-moment microphysics scheme. It turned out that by using the two-moment microphysics scheme and explicit convection best results are achieved with significant improvement towards the 28 km simulation (see attached Review-Figure-8 to Review-Figure-10, with 28 km grid mesh size (D1, blue) and 5 km (D2, green)) with Save supersite observations (black lines) as a reference.

- Review-Figure-13: Temporal evolution of the surface net downward shortwave radiation (W m-2) for the nine-day spin-up time (25 June - 3 July 2016) at Savè as observed (black, Energy Balance Station) and modeled with COSMO-ART (D1 in blue and D2 in green).

- Review-Figure-14: Temporal evolution of the 2 m temperature (°C) for the nine-day spin-up time (25 June - 3 July 2016) at Savè as observed (black, Energy Balance Station) and modeled with COSMO-ART (D1 in blue and D2 in green).

- Review-Figure-15: Temporal evolution of the 2 m relative humidity (%) for the nine-day spin-up time (25 June - 3 July 2016) at Savè as observed (black, Energy Balance Station) and modeled with COSMO-ART (D1 in blue and D2 in green).

Within this assessment, the turbulence closure had less impact on the results than the treatment of the convection. Therefore we used this 5 km COSMO-ART realization for the nesting simulations with 2,5 km grid mesh size.

(8) page 2, line 19: "react" –> "are" or other fix

**We have changed the manuscript accordingly.**

(9) page 3, line 1: "dependent" –> "dependence" or other fix

**We have changed the manuscript accordingly.**

(10) recommend to divide section 5 text up from one long paragraph currently

**We agree on that and subdivided Section 5 in five paragraphs.**

(11) recommend to guide the reader more graphically in following the transition from figure 2 (schematic diurnal cycle) to later figures (all in UTC), such as by indicating UTC time range on the panels of figure 2

**We agree on that and added the approximated UTC time range in the caption of Figure 2. Nevertheless, it has to be noted that this is just a rough estimation. Kalthoff et al. (2018) show that the onset and the evolution of the NLLS can vary considerably from day to day and from one site to an other.**

Additional References Deetz, K.: Assessing the Aerosol Impact on Southern West African Clouds and Atmospheric Dynamics, Dissertation, KIT Scientific Publishing, Karlsruhe, 75, 2018.

Deetz, K., Klose, M., Kirchner, I., and Cubasch, U.: Numerical simulation of a dust event in northeastern Germany with a new dust emission scheme in COSMO-ART, Atmos. Environ., 126, 87–97, 2016.
* * *
**Lead time (h)**

[Figure]

**Fig. 1.** Review-Figure-1: Temporal evolution of the height (m AGL) of the wind speed maximum between 0 and 1500 m AGL for the mean 57 h forecast lead time. (13 June - 30 July 2016) at Savè.

[Figure]

**Fig. 2.** Review-Figure-2: Wind speed profile (m s-1) between 0 and 2000 m ASL as mean diurnal cycle (13 June - 30 July 2016) at Savè for (a) COSMO-ART and (b) Doppler Lidar observation.

[Figure]

**Fig. 3.** Review-Figure-3: Vertical profiles (km AGL) of BC (mg m-3) at Savè for (a-c) 5 July 2016, (d-g) 14 July 2016 and (h-i) 15 July 2016.

[Figure]

**Fig. 4.** Review-Figure-4: Comparison of Twin Otter measurement flight TO-16 (14 July 2016, 06:44 UTC to 09:50 UTC) results with COSMO-ART.

[Figure]

**Fig. 5.** Review-Figure-5: Mean total AOD averaged from 27 June to 17 July 2016 of (a) COSMO-ART collocated with MODIS Terra, (b) MODIS Terra, (c) COSMO-ART, collocated spatiotemporally with

[Figure]

**Fig. 6.** Review-Figure-6: Number of observations within the time period 27 June - 17 July of (a) MODIS Terra and (b) MODIS Aqua.

**Fig. 7.** Review-Figure-7: AOD (550 nm) at Savè from COSMO-ART (blue), CAMS (green) and AERONET (red) between 13 June - 31 July 2016 for (a) mineral dust, (b) sea salt, (c) anthropogenic aerosol and (d) total

[Figure]

**Fig. 8.** Review-Figure-8: Flight track of the ATR42 SAFIRE on 3 July 2016 between 08:32 UTC and 13:13 UTC in (a) horizontal and (b) vertical dimension (m AGL). For (a) the topography (m ASL) is added. The flig

10 UTC     11 UTC     12 UTC     13 UTC

**Fig. 9.** Review-Figure-9: Aerosol number density (AND, cm-3) in the size interval 0.02 to 0.5 mu m as measured by the Spectrometer Scanning Mobility Particle Sizer (SMPS) on board the ATR42 (black) and modeled

**Fig. 10.** Review-Figure-10: Comparison between Savè supersite observations (grey) and COSMO-ART reference (black), clean (blue) and polluted (red) of (a) net downward shortwave radiation (W m-2), (b) net downwa

[Figure]

**Fig. 11.** Review-Figure-11: Temporal evolution of median (a) total aerosol number (cm-3) and (b) total aerosol mass (mu g m-3) in the lowest 2 km AGL over Ivory Coast (7.5–3.0°W) between 2 July 15 UTC to 3 Ju

[Figure]

**Fig. 12.** Review-Figure-12: Temporal evolution of the differences in surface sensible heat flux (red, W m-2), surface latent heat flux (green, W m-2) and surface pressure (blue, hPa).

[Figure]

**Fig. 13.** Review-Figure-13: Temporal evolution of the surface net downward shortwave radiation (W m-2) for the nine-day spin-up time (25 June - 3 July 2016) at Savè as observed (black, Energy Balance Station) a

[Figure]

**Fig. 14.** Review-Figure-14: Temporal evolution of the 2 m temperature ($^{\circ}$C) for the nine-day spin-up time (25 June - 3 July 2016) at Savè as observed (black, Energy Balance Station) and modeled with COSMO-ART (

[Figure]

**Fig. 15.** Review-Figure-15: Temporal evolution of the 2 m relative humidity (%) for the nine-day spin-up time (25 June - 3 July 2016) at Savè as observed (black, Energy Balance Station) and modeled with COSMO-A

---

## Author Comment (AC2) · 19 Jun 2018

Answer to Referee #2 Konrad Deetz 19 June 2018

Dear Referee (Atmospheric Chemistry and Physics),

thank you for your report from 16 April 2018. We have accounted for the comments and suggestions in the revised manuscript version. Please find our replies (marked with #) to the individual comments in the following. Before the detailed replies to your comments we want to stress one important overarching point: This study mainly focuses on the sensitivity of atmospheric dynamics and cloud properties to aerosols and not on a

detailed validation of the model system. Nevertheless, we have done a comprehensive evaluation of the model with the available observations of the DACCIWA measurement campaign. We show corresponding figures in our replies which appear at the end of this document. (The complete figure captions are given within the text because the figure caption space for the uploaded figures is not sufficient.)

Sincerely, Konrad Deetz on behalf of all coauthors

Referee comments:

(0) The authors focus on South West Africa, a region which is in a developing phase with an expected massive population growth and urbanisation. Therefore, an increase in anthropogenic aerosol concentration is expected. The authors assess the implication of aerosols and their possible changes on clouds and atmospheric dynamics. They present a process study with the regional model COSMO-ART. In particular, they discuss the impacts of aerosols on the propagation of the Atlantic Inflow frontal location and the Stratus to Cumulus Transition. In general, the paper is well written and the topic is of general interest. Deetz et al. conducted a detailed analysis of the performed simulations to understand the processes how aerosols influence prominent South West African dynamical features. Their main conclusions are based on three different simulations, the reference, the polluted and the clean case.

(1) I miss the discussion about the realistic representation of the current aerosol distribution in the model. Since an extensive measurement campaign took place during July 2016 it should be possible to evaluate the simulated distribution of aerosols against more measurements (here only a comparison with measures liquid cloud properties are shown).

**The DACCIWA observations focus more on the eastern part of SWA and not on central Ivory Coast where we set the focus due to the pronounced evolution of the AI front. The supersites are located in Ghana, Benin and Nigeria. Remote sensing aerosol measurements are impeded by clouds and the supersites does not provide aerosol information apart from sun photometer observations. The only source of aerosol observations are the research aircrafts. And these information does not suffice to get a clear picture of the current aerosol distribution at least over Ivory Coast. The assessment of the current aerosol distribution is further impeded by the overall uncertainty of the emission datasets. Therefore, we can provide and assess effects of relative changes in the aerosol amount but cannot clearly define the actual state. As mentioned in the text, the period 3-4 July was selected because of extensive NLLS at Save, standing this day out as a golden day for further research. Unfortunately, for 3-4 July no aircraft observations were available for Ivory Coast. This data shortcoming we bypassed by evaluating the model in the eastern part of the domain via aircraft observations in the Lome-Save area (Fig. 3 and 4). To meet your concerns, we added further evaluations of aerosol properties in the Lome-Save area by using observations from the ATR42 SAFIRE aircraft. However, this evaluation focuses on the Lome-Save area and not on Ivory Coast. In Section 2.2 (Observational data) we added the following text: "The aerosol aerosol number density is evaluated using observations of the ATR42 SAFIRE (Service des Avions Français 25 Instrumentés pour la Recherche en Environnement) for the 3 July 2016. Additionally, the comparison of the modeled net downward shortwave and longwave radiation as well as the sensible and latent heat flux with Savè supersite is presented in Figure 19 of Appendix B. COSMO-ART reasonably reproduces the fluxes with lower fluxes with increasing aerosol as expected." In Section 4 (Evaluation of modeled cloud and aerosol properties with aircraft observations) we added the following text: "The research aircraft ATR42 SAFIRE also obtained aerosol properties in the Lomé-Savè area on 3 July 2016 (8:32–13:16 UTC). The flight track and altitude is presented in Figure 5, showing similar flight patterns compared to the Twin Otter (Fig. 3). By assuming dry aerosol, Figure 6 shows the comparison between COSMO-ART and the Spectrometer Scanning Mobility Particle Sizer (SMPS) to evaluate the Aerosol Number Density in the size range 0.02–0.5 mu m. Figure 6 reveals that the modeled aerosol number density shows a similar temporal evolution compared to the observations but has a constant bias, overestimating the observed aerosol number**

density by a factor of about 2 (indicated by the blue dashed line). Therefore, in the subsequent study it has to be considered that the reference case shows already higher aerosol concentrations compared to the current state in SWA as quantified by the aircraft measurements. Overall, the evaluation reveals that COSMO-ART is capable to reproduce the aerosol situation on 3 July 2016 over SWA which is the basis for further sensitivity studies.

The figures related to this passage are attached in this review answer:

Fig.5 -> Review-Figure-1: Flight track of the ATR42 SAFIRE on 3 July 2016 between 08:32 UTC and 13:13 UTC in (a) horizontal and (b) vertical dimension (m AGL). For (a) the topography (m ASL) is added. The flight track in (a) and (b) is separated in hourly time steps for the subsequent collocation with hourly model data from COSMO-ART, highlighted by the pink (08:32—09:30 UTC), blue (09:30–10:30 UTC), gray (10:30–11:30 UTC), red (11:30–12:30 UTC) and black color (12:30–13:13 UTC). Furthermore, the arrows in (a) indicate the flight direction with the takeoff at Lomé, the flight to Savè and the return to Lomé airport. Shortly. Note the meridional compression of the map in (a).

Fig.6 -> Review-Figure-2: Aerosol number density (AND, cm-3) in the size interval 0.02 to 0.5 mu m as measured by the Spectrometer Scanning Mobility Particle Sizer (SMPS) on board the ATR42 (black) and modeled with COSMO-ART (solid blue, reference case). The horizontal dashed blue line shows the COSMO-ART AND divided by 2. The vertical blue dashed lines indicate the COSMO-ART model output hours, which are compared to the observations.

Fig. 19 (Appendix B) -> Review-Figure-3: Comparison between Savè supersite observations (grey) and COSMO-ART reference (black), clean (blue) and polluted (red) of (a) net downward shortwave radiation (W m-2), (b) net downward longwave radiation (W m-2), sensible heat flux (W m-2) and latent heat flux (W m-2). The horizontal lines in (a) denote clouds over Savè in the observations and COSMO-ART.

(2) Since the direct aerosol effect depends mainly on the radiative properties of the aerosols it is of interest to show the aerosol composition in the region during the 2nd -3rd of July. And again, it would be helpful if the simulated aerosol radiative properties or the simulated radiative fluxes could be evaluated against observations. To understand the full meaning of polluted and clean case it is necessary to know about the aerosol content and composition in the reference case. Without that knowledge, a fractional increase or decrease is not meaningful. Do the authors change the aerosol concentration of the different types equally? This should be clarified in the revised manuscript.

**Refers to (1). We see your point and to consider your remark we have added a comparison of net downward shortwave radiation, net downward longwave radiation, sensible heat flux and latent heat flux with respect to the supersite Save (Appendix B (Fig. 19) -> Review-Figure-3). Yes, all aerosol types are changed equally by the factor. We clarified the following sentence to make this more precise: " All aerosol modes and thus all aerosol types are changed uniformly by the factors."**

(3) Another clarification is needed in terms of the general model setup. How are aerosols treated at the outer boundaries? Are they prescribed by output of global model simulations? The meteorological state is initialized every day at 0 UTC. Are the wind and temperature fields pulled back to the ICON forecast every day at 0 UTC? If yes, how is it possible to analyse the impact of the direct and indirect aerosol effect on the dynamics? I also wonder about the choice of the inner model domain (figure 1, indicated by red box). The western as well as the eastern and part of the northern boundary are located in a mountainous region. Could that cause problems due to resolution effects?

**As denoted in Section 2.1 and Table 2, the COSMO-ART aerosol-interaction simulations (2.5 km grid mesh size) are realized via a nesting in a COSMO-ART realization with 5 km grid mesh size. For this simulation the aerosol boundary is taken from the global model simulation MOZART (see Tab. 2). Therefore, the aerosol coming from the**

boundary into the domain of the 2.5 km domain is a combination of MOZART aerosol and aerosol that is emitted within the 5 km domain. Just for clarification: The aerosol scaling is just done within the aerosol activation and the radiative transfer calculation and not in the entire aerosol dynamics. Therefore it does not matter whether the aerosol is emitted locally in the 2.5 km domain or advected from outside. No, COSMO-ART does not include a two-way nesting. Therefore there is no feedback from the 2.5 km domain to the 5 km domain. This feature will be available in the new model system ICON-ART. Also without two-way nesting, the 2.5 km domain developes its on dynamics. The predominant wind direction is southwest via the monsoon flow. So the wind is coming from the southeast Atlantic. The SST is fixed in the 2,5 km realization as well as in the coarser domain (5 km). The aerosol effect on AI and SCT is therefore only and directly evolving in the 2.5 km realization because the southern AI "boundary condition" (incoming monsoon flow) and the northern AI "boundary condition" (saharan heat low) are unaffected. We are just focusing on the changes in between and this is the 2.5 km domain. The mountains heights in the domain are below 1 km, in most cases below 500 m. With the predominant wind direction southwest we have not faced any problems.

(4) Page 1 line 22: The population is expected to growth.

**Please specify. Do you propose to replace the two sentences: "More than half of the global population growth between now and 2050 will occur in Africa. For Nigeria, which has a population of 182 million in 2015 (rank 7), a population increase to 399 million (rank 3) is expected for 2050 (UNO, 2015)." by your sentence? To highlight that especially SWA will be affected by a significant population increase which will be directly linked to a substantial enhancement of air pollution, we would like to include the population projections from UNO (2015).**

(5) Page 2 line 19: Please replace "react" with "are".

**We agree on that and have changed the manuscript and the figures accordingly.**

(6) Page 10 line 25: Please rewrite that sentence

**Please specify. We are interested in the results of LES aerosol-atmosphere inter-action simulations in the framework of DACCIWA since the aerosol effects on spatial scales of 100 m might be different to the COSMO-ART results on spatial scales of 2500 m. With this sentence we will express our interest and simultaneously provide an outlook and link to other research that is done in DACCIWA and which is related to our field of research.**

(7) Table 1: I recommend to rename the simulations (the names are unnecessary long), ADE and AIE are scaled by the same factor, the simulations could be named as AE0.1, AE0.25... (AE = aerosol effect)

**We agree on that and have changed the table as well as figure captions and legends accordingly.**

(8) General remark: Maybe it is not necessary to present results of all 6 simulations. It underlines somehow the results but for the discussion it seems not to be important to present them. I recommend that the authors rethink the demand to present the AE0.1, AE0.5, AE2 simulations in the paper.

**As you have remarked, the purpose of these additional realizations is to underline the results and to make the conclusions based on the realizations more robust. Just from three realizations it can hardly be concluded about the relationship of aerosol change and atmospheric response, e.g. whether it is linear or nonlinear. Your remark is in contrast to the remarks provided by referee #3, asking for longer simulation periods. These realizations are very expensive but we did these three additional runs and we gained added value from them. Furthermore, we are aware of the problem that showing results of all realizations might confuse the reader. Therefore only two selected figures present all realizations (Fig. 9 and 12). All other figures refer to the key realizations of clean, reference and polluted.**

[Figure]

**Fig. 1.** Review-Figure-1: Flight track of the ATR42 SAFIRE on 3 July 2016 between 08:32 UTC and 13:13 UTC in (a) horizontal and (b) vertical dimension (m AGL). For (a) the topography (m ASL) is added. The flig

10 UTC    11 UTC    12 UTC    13 UTC

AND (cm$^{-3}$)

— ATR SMPS
— COSMO−ART
- - 0.5 COSMO−ART

20000
15000
10000
5000
0

10:00    11:00    12:00    13:00

**Time (UTC)**

**Fig. 2.** Review-Figure-2: Aerosol number density (AND, cm-3) in the size interval 0.02 to 0.5 mu m as measured by the Spectrometer Scanning Mobility Particle Sizer (SMPS) on board the ATR42 (black) and modeled

[Figure]

**Fig. 3.** Review-Figure-3: Comparison between Savè supersite observations (grey) and COSMO-ART reference (black), clean (blue) and polluted (red) of (a) net downward shortwave radiation (W m-2), (b) net downwar

---

## Author Comment (AC3) · 19 Jun 2018

Answer to Referee #3 Konrad Deetz 19 June 2018

Dear Referee (Atmospheric Chemistry and Physics),

thank you for your report from 17 April 2018. We have accounted for the comments and suggestions in the revised manuscript version. Please find our replies (marked with #) to the individual comments in the following. Before the detailed replies to your comments we want to stress one important overarching point: This study mainly focuses on the sensitivity of atmospheric dynamics and cloud properties to aerosols and not on a

detailed validation of the model system. Nevertheless, we have done a comprehensive evaluation of the model with the available observations of the DACCIWA measurement campaign. We show corresponding figures in our replies.

Sincerely, Konrad Deetz on behalf of all coauthors

Referee comments:

(0) In this study, the authors ran a regional climate model COSMO-ART at convection-perming resolutions to examine the effects of aerosols on weather and climate based on a 2-day case study. They documented in detail how aerosol affects cloud and atmospheric dynamics over southern West Africa. They further presented detailed analysis of mechanisms that leads to these changes, and provide a conceptual model for this. I think overall the paper is well written, and it is great addition to existing literatures on aerosols effects on climate over Africa. I would recommend it publication after my following comments are addressed:

(1) I understand these are expensive simulations, but I still think it would be really nice if the authors can run model longer, say a month. The current results are interesting, but it is less clear how robust these results are. A longer simulation would definitely be more interesting, and may also produce more robust results.

**We see your point. It would be great to have model results for the entire monsoon post-onset phase (22 June to 20 July), but as you said, the realizations are expensive, expensive with respect to computing time and also of handling the very large amounts of data. In terms of resources it is simply not feasible for us to realize this. Therefore, and after due consideration we decided to focus on the 3-4 July 2016, identified by the DACCIWA community as a golden day for further research. As denoted in the manuscript, from our experience during the measurement campaign, there is very small variation in the general meteorological conditions during the monsoon post-onset phase. Therefore we can assume well grounded, that our sample '3-4 July' within the post-onset phase is representative at least qualitatively for the entire post-onset phase**

(nearly one month) without simulating the whole period. In our companion paper about the aerosol liquid water content in SWA, submitted on 26 April 2018 to ACP (also DAC-CIWA special issue) and which is likely soon also in the discussion phase, we added results of the 6-7 July in the Appendix (as a second sample within the monsoon post-onset phase), underlining that the conditions are similar from one day to another.

(2) Their model does have the capability to separately treat AIE and ADE. But in the paper, the authors examined the two effects together. Separating these two may help to answer whether AIE or ADE dominates in this case study.

**The decision of using ADE and AIE together as well as using the same factor for both, within one realization, is made after due consideration. Generally, it is possible to have a realization just with AIE (ADE turned off). On the other hand it is not appropriate to have a realization just with ADE since you cannot simply switch off AIE. In any case you have to have aerosol as CCN. If you switch AIE off, you will use the aerosol climatology given in the two-moment scheme of COSMO instead. Simultaneously, in ADE you use the prognostic aerosol of COSMO-ART. This is totally inconsistent and does not allow any conclusions about the single effects of ADE and AIE. The alternative is to use the factors to change the aerosol that is seen by ADE or AIE, as it is done in this study. With this you can reduce e.g. AIE but still (consistently) the same aerosol description (COSMO-ART prognostic) is the basis for AIE and ADE. Now, the question about turning off one of the effects completely changed to the question of using different or the same factors within one realization. We have tested it to gain knowledge by using e.g. ADE0.1-AIE1.0. But again, this is physically inconsistent and cannot be interpreted in terms of realistic conditions in SWA. E.g. in ADE0.1-AIE1.0 the incoming solar radiation would be higher than it would appear in reality (less scattering and absorption), leading very likely to more convective processes (especially sensitive in tropical regions) and therefore implications on AIE that would not occur in the realistic setup ADE1.0-AIE1.0. Using different factors is only appropriate when using the factorialmethod proposed by Montgomery (2015). In my dissertation (Deetz,**

2018) I used the factorial method to assess whether the AI frontal shift is caused by AIE or ADE. The findings underline that ADE is dominating the changes, nevertheless we decided not to have this method included in this publication because for robust results an ensemble approach is necessary as realized in Kraut (2015). To have one realization only with AIE (a) and one realization with AIE and ADE together (b) and assessing the effect of ADE by calculating (b) minus (a) is also problematic because this would not only include effects from ADE but also synergistic effects of AIE and ADE. By having the same factor for both effects, every realization is in itself consistent and allows for meaningful, physically-reasonable conclusions. Therefore we decided to use ADE and AIE together and with the same factors.

(3) Title: The paper is about aerosol effects on atmospheric dynamics in a case study. But the title said "cloud and aerosol radiative effects as key players for anthropogenic changes in atmospheric dynamics over southern West Africa". I think the title is misleading and confusing. FirstïijŃËĞ the paper is not about cloud radiative effectsïijŃ ËĞ though it does talk about aerosol radiative effects through its impact on clouds. But this is different from cloud radiatve effects. Second, the paper only documents aerosol effects on atmospheric dynamics based a case study from model simulations. "anthropogenic changes in atmospheric dynamics" may sound like this is what you observed. As this effect is purely a modeling study, I suggest the authors to clarify this in the title.

**We rephrased the title: "Numerical simulations of aerosol radiative effects and their impact on clouds and atmospheric dynamics over southern West Africa"**

(4) Section 2.1: model experiments and AIE. It looks like the authors can separately examine the effects of AIE and ADE, but in all model experiments documented here, AIE and ADE are examined together. If the authors examine AIE and ADE separately, this may help to clarify some points the authors made regarding the relative roles of AIE and ADE on SWA. This relates to some of the discussions in Section 6 (e.g., the last paragraph).

\# Refers to (2).

(5) Section 5: the first paragraph is overly long. Suggest to separate it into several short paragraphs with a focus theme in individual paragraphs.

\# We agree on that and have changed the manuscript accordingly.

(6) Page 9, line 21: what are these two numbers? The same question is also applied for next three lines (lines 22-24)

\# As denoted on page 9, line 18-19: "The following values in brackets indicate the median and the 99 th/1th percentile of the surface quantities considering the cloud-free inland area." With this we want to provide quantitative expressions of the observed changes that are consistent between the different meteorological parameters. We decided to present the median and the 99th percentile (in case an increase is observed) or 1th percentile (in case a decrease is observed). This is also consistent for the values in the next three lines.

(7) Page 10, line 4-5: an aerosol increase has large impacts than the aerosol decrease. This is a little bit surprise to me. I would expect when aerosol concentrations further increases, its effects saturate, and its effects decreases (e.g., numerous small particles compete for water vapor so a lower maximum supersaturation is expected). So can you elaborate what might happen here.

\# Figure 12 (old manuscript version) / Figure 14 (new manuscript version) shows that we have not reached the saturation point. Moreover the strong increase of the CDNC coincides with decrease in the cloud water content, a decrease in the effective radius and a decrease (increase) in light (heavy) precipitation in agreement with the convective invigoration mechanisms of warm clouds as described in Saleeby et al. (2014).

(8) Page 10, lines 12-14: The Twomey effect is also through changes in cloud optical thickness, but not through cloud water. So the second half of this statement is confusing.

\# We rephrased this sentence: Figure 12 reveals that the aerosol impact on radiation via the Twomey effect is very likely dominating the cloud-radiation interaction, whereas changes in the cloud water, that can also impact the radiative transfer and therefore the cloud formation, are insignificant.

(9) Page 10, lines 28-29: this statement is not clear to me ("it is interesting that ...").

\# "Nevertheless, it is interesting that the location of the AI front during its stationary phase over Ivory Coast could be used as a proxy for the aerosol burden in that area (under otherwise identical conditions)." The balance between the onshore monsoon flow and the vertical mixture of momentum over land due to turbulence leads to stationarity of the AI front around noon. The higher the aerosol burden, the smaller the vertical mixture of momentum over land and the more the onshore monsoon flow dominates in the denoted balance. Therefore the AI front is shifted inland but is still stationary. With that, the location of the AI front around noon, relative to the coast, is a proxy for the aerosol burden over land (at least in the model under otherwise same conditions).

(10) Page 10, the pressure gradient mechanisms: Here sea surface temperature was not affected by aerosol loading. So this overestimates the effects of aerosols on land-sea temperature differences. Any discussion on this?

\# Theoretically, we can expect a decrease in SST with increasing aerosol due to less incoming shortwave radiation via scattering and absorption. This SST decrease might lead to a decrease of the near-surface temperature over the ocean. (The near-surface temperature decrease over land due to the aerosol is actually happening in the model.) All in all, this would lead to a stronger land-sea temperature gradient and therefore to a less weak AI inland propagation compared to the results of the polluted case in the manuscript. To study aerosol effects on the SST and its feedbacks on AI, the NWP mode of COSMO is not appropriate. For that, climate studies with the CLM version of COSMO are more meaningful. However, we don't expect significant signals on SST, because on the one hand the surface water is continously moved by the Benguela

current and Southern Equatorial current and on the other hand, the biomass burning aerosol is reaching the Gulf of Guinea not continously but it arriving in plumes as visible e.g. in model realization of COSMO-ART and MACC. Therefore there will be no continuous cooling but a variation of cooled and warmed areas that might counterbalance. Furthermore, a regional (limited-area) model is less appropriate for studying this question, because only the near-surface water within the domain is affected but the incoming water is unaffected. Therefore a global model, consistently having the aersol impact on SST globally considered, would be more appropriate for this questions. We added the following sentence in the conclusions: "Effects on SST are not considered in this study. In case of considering the impact of reduced incoming solar radiation on the SST with increased aerosol, stronger land-sea temperature gradients are expected. Therefore, the estimations of this study with fixed SST denote the upper limit of the magnitude of the effects. However, this model setup in numerical weather prediction mode is less appropriate to study effects on SST. Global models on a longer time scale are more suitable to provide added value on this question."

Additional References Deetz, K.: Assessing the Aerosol Impact on Southern West African Clouds and Atmospheric Dynamics, Dissertation, Wissenschaftliche Berichte des Instituts für Meteorologie und Klimaforschung des Karlsruher Instituts für Technologie, KIT Scientific Publishing, Karlsruhe, 75, 171-172, 2018.

Kraut, I.: Separating the Aerosol Effect in Case of a "Medicane", Wissenschaftliche Berichte des Instituts für Meteorologie und Klimaforschung des Karlsruher Instituts für Technologie, Dissertation, 2015.

Montgomery, D. C.: Design and Analysis of Experiments, 5th ed., John Wiley, New York, 115, 684–686, 2005.